

# Effect of disdrometer type on rain drop size distribution characterisation: a new dataset for Southeastern Australia

Adrien Guyot[1], Jayaram Pudashine[1], Alain Protat[2], Remko Uijlenhoet[3], Valentijn R.N. Pauwels[1], Alan Seed[2] and Jeffrey P. Walker[1]

[1] Department of Civil Engineering, Monash University, Melbourne, Victoria, Australia

[2] Bureau of Meteorology, Melbourne, Victoria, Australia

[3] Wageningen University, Hydrology and Quantitative Water Management Group, Wageningen, The Netherlands

*Correspondence to*: Adrien Guyot (adrien.guyot@monash.edu)

**Abstract.** Knowledge of the full rainfall Drop Size Distribution (DSD) is critical for characterising liquid water precipitation for applications such as rainfall retrievals using electromagnetic signals and atmospheric model parameterisation. Southern Hemisphere temperate latitudes have a lack of DSD observations and their integrated variables. Laser-based disdrometers

rely on the attenuation of a beam by falling particles and is currently the most commonly used type of instrument to observe the DSD. However, there remain questions on the accuracy and variability in the DSDs measured by co-located instruments wether identical models, different models or from different manufacturers. In this study, raw and processed DSD observations obtained from two of the most commonly deployed laser disdrometers, namely the Parsivel[1] from OTT and the Laser Precipitation Monitor (LPM) from Thies Clima, are analysed and compared. Four co-located instruments of each type

were deployed over 3 years from 2014 to 2017 in the proximity of Melbourne, a region prone to coastal rainfall in Southeast Australia. This dataset includes a total of approximately 1.5 million recorded minutes, including over 40,000 minutes of quality rainfall data common to all instruments, equivalent to a cumulative amount of rainfall ranging from 1093 to 1244 mm (depending on the instrument records) for a total of 318 rainfall events. Most of the events lasted between 20 and 40 min for rainfall amounts of 0.12 mm to 26.0 mm. The co-located LPM sensors show very similar observations while the co-located

Parsivel[1] systems show significantly different results. The LPM recorded one to two orders of magnitude more smaller droplets for drop diameters below 0.6 mm compared to the Parsivel[1], with differences increasing at higher rainfall rates. The LPM integrated variables showed systematically lower values compared to the Parsivel[1]. Radar reflectivity-rainfall rate ($Z_H$-R) relationships and resulting potential errors are also presented. Specific $Z_H$-R relations for drizzle and convective rainfall are also derived based on DSD collected for each instrument type. Variability of the DSD as observed by co-located

instruments of the same manufacturer had little impact on the estimated $Z_H$-R relationships for stratiform rainfall, but differs when considering convective rainfall relations or $Z_H$-R relations fitted to all available data. Conversely, disdrometer-derived $Z_H$-R relations as compared to the Marshall-Palmer relation $Z_H = 200R^{1.6}$ led to a bias in rainfall rates for reflectivities of 50





dBZ of up to 21.6 mm h⁻¹. This study provides an open-source high-resolution dataset of co-located DSD to further explore sampling effects at micro-scale, along with rainfall microphysics.

**Keywords:** Disdrometers, Precipitation, Weather Radar, Rain Drop Size Distribution, Rainfall, Reflectivity.







## 1 Introduction


Detailed knowledge of local rain microphysics is important for a range of applications: to better understand hydro-meteorological regimes and climate characteristics of a specific region, to study interactions between atmospheric and land surface processes, but also to determine characteristics of cloud and precipitation formation. Since the advent of weather radar and associated rainfall nowcasting applications, quantitative knowledge of the rain microstructure and knowledge of

hydrometeor size distributions within precipitating clouds has become of great importance in order to relate backscattered radar signals to quantitative precipitation amounts (Marshall et al., 1955; Uijlenhoet, 2001). More broadly, the energy in the form of microwaves travelling through the atmosphere is susceptible to attenuation by vapor or liquid water. A quantitative knowledge of rain microphysics is thus critical to understand and predict how signal propagation is altered by precipitation. The data sources include weather radars, radiometers, and microwave communications at ground level, in particular

Commercial Microwave Links (CML). Relations between Quantitative Precipitation Estimates (QPE) and microwave signal (back)scattering, *e.g.* reflectivity and attenuation, highly depend on the rain microphysics, and accurate QPE therefore requires detailed knowledge of the rain microstructure (Adirosi et al., 2018).

Rain microphysical properties are typically characterized by particle or drop size distribution (PSD or DSD) and/or particle size and velocity distribution (PSVD). These are not routinely measured *in-situ* in the cloud or aloft, but at ground level,

where long-term deployments of observational instruments is feasible. Historical observations have relied on manual sampling including stain and oil immersion techniques. Automatic disdrometers have since been designed to measure PSD and/or PSVD using either mechanical impact principles, where the falling drops hit a pressure sensor which converts this into an electrical current (Joss and Waldvogel, 1967), or the laser-extinction principle (Illingworth and Stevens, 1987) whereby the falling particles modify the laser beam between an emitter and receiver from which the PSVD is derived, and

finally a video-based principle (Kruger and Krajewski, 2002; Schonhuber et al., 2008), where a combination of high-speed line-scan cameras record the PSVD.

Each method and corresponding hardware varieties have their advantages and drawbacks, leading to different uncertainties and errors in the measured PSVD and PSD. Typically, video disdrometers (two-dimensional video disdrometers or 2DVD) were initially considered the most reliable to accurately measure PSVD (Thurai et al., 2017), particularly for particles larger

than 0.3 mm. Recent work has re-visited this and now consider the 2DVD to significantly underestimate small particles in particular (Thurai and Bringi, 2018). Long-term deployments are often difficult and cost-prohibitive and as a result these 2DVD instruments have typically been deployed for short-term research experiments. Thurai et al. (2017) presented data from a Meteorological Particle Spectrometer (Baumgardner et al., 2002), arguing its higher sensitivity and better accuracy for diameters below 1.1 mm as compared to the 2DVD, while the 2DVD was proven to be accurate above 0.7 mm. The most

commonly used disdrometer types at present are the laser-based disdrometers (Kathiravelu et al., 2016; Angulo-Martinez et al., 2018) with only a handful of manufacturers commercializing such instruments. The most commonly used disdrometers,



with the highest citations in the scientific literature, are the particle size and velocity Parsivel of the first and second (released in 2011) generations by OTT Hydromet. Another widely used disdrometer, based on the exact same principle but with differences in the hardware design and internal processing, is the Laser Precipitation Monitor (LPM) by Thies Clima

(De Moraes et al., 2011; Sarkar et al., 2015). These are predominantly used in research settings, and sometimes in operations where the only sought-after information is a precise and accurate estimate of the rainfall amount (Merenti‑Välimäki et al., 2001).

Since all PSVD observations using sensing techniques are subject to biases and errors, one of the most relevant approaches to accurately determine DSD is to use co-located instruments, preferably of different types and brands to compare and

evaluate the estimates from different instrumental sources. Such approaches have been used to evaluate DSD measured simultaneously by Parsivel[1]-only (Tapiador et al., 2010; Jaffrain and Berne, 2011; Jaffrain et al., 2011; Jaffrain and Berne, 2012), Parsivel and 2DVD (Raupach and Berne, 2015), Meteorological Particle Spectrometer (or Sensor) and 2DVD (Thurai et al., 2017; Thurai and Bringi, 2018) and Parsivel[2] and LPM (Angulo-Martinez et al., 2018). Work assessing the accuracy and biases of Parsivel has been widespread, while evaluation of the Thies LPM has received comparatively little attention

(Brawn and Upton, 2008; Upton and Brown, 2008; Adirosi et al., 2018; Angulo-Martinez et al., 2018). Detection ranges vary slightly between Parsivel and LPM, with the LPM having a lower particle size detection threshold. Angulo-Martinez et al. (2018) filtered the raw matrices to a common detection range to allow direct comparison. Nevertheless, they found significant differences between the disdrometer models for both the PSVD and the integrated DSD moments.

In Australia, DSD observations have been conducted for the tropical region near Darwin over the past 20 years. Furthermore,

extensive campaigns have been conducted to monitor PSVD (Maki et al., 2001; Penide et al., 2013; Thurai et al., 2010), and recent dedicated research vessel campaigns over multiple years have led to characterization of the Australian sector of the Southern Ocean, which influences Australian weather and climate (between latitudes of 40° and 65° South; Mace and Protat, 2018ab). PSVD were measured using optical disdrometers specifically designed for harsh environments (Klepp et al., 2018). However, a gap of knowledge remains for the temperate mid-latitudes of Australia, which also seems to be the case for the

entire Southern Hemisphere mid-latitudes. In Australia, more than 80% of the population lives in the main metropolitan areas that are located along the south-eastern coastline (Sarkar et al., 2018). This region is therefore of particular interest for improved rainfall estimation using remote sensing techniques such as weather radar, CML or satellite, which all rely on accurate parameterizations of the DSD properties. The objectives of this study are therefore to: (1) evaluate the differences between raw and processed PSVD, DSD and derived integrated variables from two laser disdrometer types: the OTT

Parsivel[1] and the Clima Thies LPM; (2) provide a comprehensive quantitative description of the DSD for the Melbourne region and climate; and (3) develop reflectivity – rainfall rate and attenuation – rainfall rate relationships for this region. To achieve this, four co-located laser-based disdrometers of two manufacturers (OTT and Thies) were deployed over a three-year period in Melbourne, Australia.



## 2. Instruments and methods

**2.1 Experimental site and regional setup**

The observational site was located at the Australian Bureau of Meteorology experimental research and educational site at Broadmeadows (Figure 1), a Metropolitan suburb located north of Melbourne, Australia (37° 41' 27.4668" S, 144° 56' 54.186" E). The site is situated approximately 15 km from the Port Phillip Bay shoreline and 85 km from the Bass Strait ocean shoreline. The Melbourne metropolitan area is classified as a marine west coast climate (Cfb, Köppen-Geiger

classification). The average cumulative rainfall precipitation recorded at Melbourne Airport (weather station 086282 operated by the Bureau of Meteorology), located 9 km West from the experimental site, was 535 mm for the 1970-2018 period with a monthly maximum in November of 62 mm and a monthly minimum in July of 36 mm. However, there is relatively little variation in the monthly rainfall amounts throughout the year. This coastal area, while being partly within the rain shadow of the Otway ranges to the Southeast, which reduces total precipitation amounts when coastal systems originate

from the Southern Ocean, is typical of the Southeast coastal climate of Australia, with most precipitation originating from the sea (Bass Strait and the Southern Ocean).

Six disdrometers were originally installed at the site (Figure 1) in July 2014, three Thies Clima LPM and three OTT Parsivel[1], but only four instruments (two LPM and two Parsivel[1]) operated continuously from July 2014 to July 2017. The other two instruments were relocated soon after the initial installation to other sites. All the instruments were installed on

individual masts separated by 1.3 m and at a height of 1.5 m above ground level (Figure 1). The laser beams of the sensors were all oriented in the same direction with raw 1-min data collected.






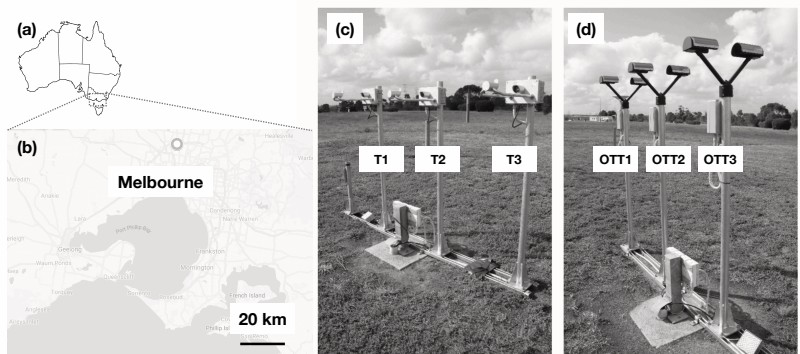

**Figure 1: (a) and (b)** Location of the Melbourne metropolitan area within Australia (source: Google Inc.) and the BoM experimental site at Broadmeadows (full circle); Picture showing the disdrometers mounted on their stands at 2 m above the ground, **(c)** Thies Clima LPM (T1, T2, T3) and **(d)** OTT Parsivel[1] (OTT1, OTT2 and OTT3).


Both instrument types are based on a similar physical principle: the attenuation of the signal strength of an infrared laser when particles pass through the light beam. Both LPM and Parsivel[1] consist of an emitter and a receiver (De Moraes Frasson et al., 2011) sampling an area of 45 to 55 cm$^2$. In both cases, some processing is done within the internal parts of the units and the "raw" data are in fact already pre-processed, accounting for some corrections applied by the proprietary software.

The "raw" data consists, for each sampling interval (in our case 1-min), of a two-dimensional PSVD matrix where the first dimension is the diameter class and the second dimension is the velocity class. The number of particles falling into each combination of diameters and velocities is counted and recorded for each time interval. Integrated variables or "moments" are also computed from the PSVD matrix by the instruments internal software and recorded for each time-step, namely the rainfall amount ($\sum R$, mm), rainfall intensity ($R$, mm h$^{-1}$), horizontal radar reflectivity ($Z_h$, dBZ), total number of detected

particles ($N_t$, # min$^{-1}$) and visibility ($Vis$, m) (Table 1). In addition, synoptic weather codes defined by the World Meteorological Organization are also attributed to each 1-min DSD following software algorithms implemented by each manufacturer.

## 2.2 Parsivel[1] from OTT

The Parsivel[1] (Figure 1) (OTT Hydromet Inc., USA) is made of two heads each mounted at the end of a V-shape, with the

emitter and receiver being slightly protected at the source by a prominent splash protection shield. Parsivel[1] uses a 650 nm laser beam generated by diodes covering an area of 54 cm$^2$, corresponding to a distance between emitter and receiver of 180 mm and a beam width of 30 mm. The minimum sensitivity of the Parsivel[1] corresponds to a particle size of 0.2 mm in



diameter. The measured voltage drops when particles cross the beam are converted into a 32 by 32 PSVD matrix with uneven bin sizes as described in Appendix 1. The first two bin sizes for particle diameters (0-0.125 and 0.125-0.250) are

systematically left empty. More details on the measurement procedure are described in Battaglia et al. (2010), Jaffrain et al. (2011), Tokay et al. (2013) and Angulo-Martinez et al. (2018). It is worth noting that a new version of the instrument was released by OTT in 2011 (Parsivel[2]), which incorporates some modifications from the version described herein (Tokay et al., 2014), including sensitivity to drop sizes in the lower and upper ranges. Specifically, Tokay et al. (2014) found the Parsivel[2] to record more droplets in the three first measurable size bins as compared to Parsivel[1], and fewer large drops as compared to

Parsivel[1]. However, issues were identified on the recorded PSVD with underestimated velocities recorded by Parsivel[2] while no issues were observed for Parsivel[1].

### 2.3 Laser Precipitation Monitor from Thies Clima

The Laser Precipitation Monitor (LPM) (Figure 1) (Thies Clima, Adolf Thies GmbH, Germany) is made of a central unit with the emitter diodes and a receiver forming an O-shape with brackets on each side of the beam. Thies LPM uses a 785 nm

laser beam covering an area of 45.6 cm[2], corresponding to a distance between transmitter and receiver of 228 mm and a beam width of 20 mm. Similarly to the OTT Parsivel[1], when a particle falls through the light beam, received signal strength is reduced and from that and the duration of the drop of signal level, particle number, sizes and their velocities are deduced for each time-step using a proprietary software. The minimum sensitivity of the Thies LPM corresponds to a particle size of 0.16 mm in diameter. Measured voltage drops when particles cross the beam, are converted into a 22 by 20 PSVD matrix

with uneven bin sizes as described in Appendix 1. A number of quality flags are also provided for each time-step to describe recorded PSVD and secondary derived data (or moments) quality.

### 2.3 Data processing

The one-minute time-step data were stored on a laptop located in a nearby building from the experimental site. Each of the instruments sent data "telegrams" after each completed min log, in the form of ASCII text data, which were then stored

within the custom software on the PC. Generation of a new log file for each sensor occurred at midnight (local PC time). The PC was connected to the Internet, therefore allowing synchronization of the internal clock. Synchronization of the disdrometers was done every day using the PC to send "telegrams" to the units for time adjustments, in order to avoid time drifts. Data generated on daily basis included raw PSVD matrices, additional derived moments and flags through various error codes.

**Table 1: List of variables, acronyms and units.**

| Variable name | Acronym | Units |
|---|---|---|
| Particle Size and Velocity Distribution | PSVD | 10*log10 (#) |





| Rainfall amount | $\sum R$ | mm |
|---|---|---|
| Rainfall intensity or rain rate | $R$ | mm h$^{-1}$ |
| Total number of detected particles | $N_t$ | unitless |
| Visibility | $Vis$ | m |
| Radar reflectivity (using the moments method) | $Z_{mom}$ | dBZ |
| Radar reflectivity derived from T-Matrix | $Z_H$ | dBZ |
| Intercept parameter of the Gamma distribution | $N_W$ | m$^3$ mm$^{-1}$ |
| Mass weighted mean diameter | $D_m$ | mm |
| Median volume diameter | $D_0$ | mm |
| Rainwater content | $W$ | g m$^{-3}$ |
| Velocity | $V_i$ | m s$^{-1}$ |
| Diameter | $D_i$ | mm |
| Time interval | $t$ | min |
| Sampling area | $A$ | cm$^2$ |
| Specific attenuation derived from T-Matrix | $\gamma$ | dB km$^{-1}$ |
| Mass spectrum standard deviation | $\sigma_m$ | mm |
| Shape parameter of the Gamma Distribution | $\mu$ | unitless |

Post-processing of the data was done using a pipeline written in the python language calling existing libraries such as pyDSD and pyTmatrix (see section on software and model codes). First a series of filtering procedures was applied: (i) data corresponding to error flags were discarded; (ii) only data corresponding to WMO synoptic weather codes 4677 and 4680 for rainfall (Drizzle, Rain and Drizzle, Rain) were retained; (iii) following Jaffrain and Berne (2011), PSVD data needed to fall between +- 50% of the Atlas et al. (1973) drop velocity model to be retained; For the Thies LPM instruments, the upper bin class for velocities (> 10 m s$^{-1}$) have no upper limit but these data were retained; (iv) minute data must meet the occurrence of more than 10 particles in three different bins (Jaffrain and Berne, 2011; Tokay et al., 2013); (v) only rainfall rates > 0.1 mm h$^{-1}$ were considered for further analysis; and (vi) contrary to Angulo-Martinez et al. (2018), the first bin diameter size for the LPM was used as, ultimately, users of this instrument will consider all available data to derive integrated DSD variables. The sensitivity of the derived integrated variables to the smaller bin sizes were tested by computing DSD integrated variables considering either the full spectra or the DSD spectra for diameters > 0.6 mm. Table 2 summarizes the statistics of the raw data and the data after applying the above successive filtering steps. Integrated variables provided by the instrument internal software were only used for reference with integrated variables or moments computed from the filtered PSVD matrices. The Drop Size Distributions were calculated using:



$$N(D_i) = \sum_{j=1}^{nd} \frac{n_{ij}}{A_i \, \Delta t \, V_j \, \Delta D_i}, \tag{1}$$

where $D_i$ (mm) is the mean volume-equivalent diameter of the $i^{th}$ bin, $N(D_i)$ (m$^{-3}$mm$^{-1}$) is the concentration of raindrops per unit volume in the interval from $D_i$ to $D_i + \Delta D_i$, $n_{ij}$ is the number of droplets recorded for measured fall speed $V_j$ (m s$^{-1}$) for velocity bin $j$, $nd$ is the number of bins for velocities (32 for OTT and 20 for Thies), $A_i$ (m$^2$) is the effective sampling area for

the $i^{th}$ size bin and $\Delta t$ (s) is the sampling interval, equivalent to 60 seconds in this study.

Both the moments and T-matrix approaches were used to compute integrated variables to describe the microphysical properties of the sampled rainfall volumes, with the radar reflectivity factor $Z_{mom}$ (dBZ) following:

$$Z_{mom} = \sum_{i=1}^{nd} \sum_{j=1}^{nv} D_i^6 \frac{n_{ij}}{A_i \, \Delta t \, V_j}. \tag{2}$$

Rain rate $R$ (mm h$^{-1}$) was calculated as:

$$R = 6\pi \, 10^{-4} \sum_{i=1}^{nd} \sum_{j=1}^{nv} D_i^3 \frac{n_{ij}}{A_i \, \Delta t}, \tag{3}$$

with the generalized intercept parameter $N_W$ (m$^{-3}$ mm$^{-1}$) (or N$_0$* as defined by Testud et al., 2001) computed from:

$$N_W = \frac{4^4}{\pi \rho_W} \left[ \frac{10^3 W}{D_m^4} \right], \tag{4}$$

where $\rho_w$ is the water density in g cm$^{-3}$ and $W$ is the rainwater content (g m$^{-3}$) derived from:

$$W = \frac{\pi}{6} \, 10^{-3} \rho_W \sum_{i=1}^{nd} \sum_{j=1}^{nv} D_i^3 \frac{n_{ij}}{A_i \, \Delta t \, V_j}. \tag{5}$$

The mass-weighted mean diameter $D_m$ (mm) is finally calculated from:





$$D_m = \frac{\sum_{i=1}^{nd} N(D_i)D_i^4 \Delta D_i}{\sum_{i=1}^{nd} N(D_i)D_i^3 \Delta D_i}. \tag{5}$$

Surface horizontal reflectivity ($Z_H$, dBZ) at S-Band was derived from the PSVD matrices using a python implementation (pyTmatrix) of the T-Matrix approach (Leinonen, 2014). The default parameters including a canting angle of 12°, the drop

shape model of Brandes et al. (2002) and a temperature of 20 °C were used for the T-Matrix calculations. $Z_H$-R power relations were then calculated using an exponential power law fit based upon a Levenberg-Marquardt minimization (More, 1978).

Horizontal ($\gamma_H$, dB km$^{-1}$) and vertical ($\gamma_V$, dB km$^{-1}$) attenuations were also calculated using the same approach (Leinonen, 2014) using the same drop shape model and parameters. Both horizontal and vertical attenuations were calculated over the

range of 0 to 70 GHz at intervals of 2 GHz. For each frequency, coefficients of the rainfall – attenuation (R = a $\gamma^b$), or attenuation – rainfall ($\gamma$ = k R$^\alpha$), power relations were derived using an exponential power law fit based upon a Levenberg-Marquardt minimization (More, 1978). These were then compared with the International Telecommunication Union attenuation model for rainfall (ITU, 2005).

Each one-minute time-step was classified as convective, stratiform or intermediate, following Islam et al. (2012). To

examine the properties of the PSD for a range of rainfall regimes, each dataset was divided into five rain-rate classes: (a) 0.1 – 2 mm h$^{-1}$; (b) 2 – 5 mm h$^{-1}$; (c) 5 – 10 mm h$^{-1}$; (d) 10 – 25 mm h$^{-1}$; and (e) > 25 mm h$^{-1}$.

Observed DSD were fitted using the Gamma distribution (Ulbrich, 1983):

$$N(D) = N_0 D^\mu \exp(-\Lambda D), \tag{6}$$

where $D$ (mm) is the particle diameter, $N_0$ (m$^{-3}$mm$^{-1-\mu}$) is the intercept parameter, $\mu$ the shape parameter and $\Lambda$ the slope parameter. The list of parameters, units and abbreviations used in the equations are listed in Table 1.


### 2.5 Auxiliary measurements

The closest tipping bucket rain gauge operating over that period was located at Essendon Airport (Bureau of Meteorology station #086038) being 5.6 km (37° 43' 26.7708" S, 144° 54' 21.7188" E) from the experimental site (Figure 1). This data was used to compare with annual cumulative amounts, but given the distance has not been used for more quantitative

analysis. Data from these two gauges were not filtered based on the disdrometers quality checks (given their distance from





the experimental site) and therefore would likely record more total rainfall for the same period. Rainfall events were identified as meeting the criteria of no rainfall recorded for 30 min before the beginning and for 30 min after the end of the event.

**3 Results**


**3.1 Rainfall climatology for the 2014-2017 period**

Table 2 presents relevant statistics that describe the full dataset obtained over three years for the four instruments. Approximately 1.5 million minutes of DSD were recorded by each of the instruments, the Thies LPM recording sensibly
more minutes than the OTT Parsivel[1] due to slight variations in the timing of installation and dismantling of the sensors. No error flags were observed for the OTT Parsivel[1] while the Thies LPM recorded a number of erroneous data. The recorded combined rainfall, rainfall + drizzle and drizzle minutes, which were selected from the total minutes corresponding to liquid precipitation, accounted for approximately 6 to 8% of the total duration of the experiment. A variety of precipitation types were observed at the site, including melting snow and hail, but these were anecdotal events covering hundreds of minutes
only.

In this study, common minutes with rainfall rates above $> 0.1$ mm h[-1] measured by all four instruments were selected for analysis, following an approach similar to Angulo-Martinez et al. (2018). This corresponded to a total of 40,062 common quality minutes across the four instruments, with cumulative rainfall ranging from 1093 to 1244 mm (depending on sensor)
over the observational period. The two Thies LPM systems recorded very similar rainfall totals, while the two OTT Parsivel[1] showed a difference of up to 100 mm during the common observational period. Overall, the OTT Parsivel[1] sensors always recorded more rainfall than the Thies LPM, despite their lower sensitivity (Figure 2). The total amounts recorded by the operational tipping bucket rain gauges located 5.6 km and 9.0 km from the disdrometers showed good agreement with the disdrometers records, with higher cumulative totals linked to the fact that these gauges were not filtered based on the
disdrometers quality checks.

**Table 2:** Summary of relevant statistics for the full dataset for the four co-located instruments.

|  | T1 | T3 | OTT1 | OTT3 |
|---|---|---|---|---|
| **Total minutes [min]** | 1,463,442 | 1,593,855 | 1,427,148 | 1,424,512 |
| **Total minutes without Error flags [min]** | 1,413,382 | 1,509,841 | 1,427,148 | 1,424,512 |
| **Rain, Rain + Drizzle, Drizzle minutes (based on weather code WMO 4677) [min]** | 99,565 | 122,165 | 123,130 | 105,701 |
| **Total Rain, Rain + Drizzle amount (unfiltered) [mm]** | 1,548 | 2,841 | 1,692 | 1,688 |
| **% Rain, Rain + Drizzle minutes [%]** | 6.93 | 7.99 | 6.96 | 6.09 |





| | | | | |
|---|---|---|---|---|
| **High quality Rain, Rain + Drizzle minutes [min]** | 93,419 | 93,419 | 58,056 | 53,381 |
| **Common, high quality Rain, Rain + Drizzle minutes [min]** | 53,472 | 53,472 | 50,430 | 50,430 |
| **Common Rain + Drizzle minutes > 0.1 mm/h [min]** | 40,062 | 40,062 | 40,062 | 40,062 |
| **Equivalent, filtered Rain, Rain + Drizzle amounts > 0.1 mm/h [mm]** | 1,093 | 1,099 | 1,244 | 1,155 |
| **% Rain minutes in winter [%]** | 33.6 | 33.6 | 33.6 | 33.6 |
| **% Rain minutes in spring [%]** | 20.6 | 20.6 | 20.6 | 20.6 |
| **% Rain minutes in summer [%]** | 15.9 | 15.9 | 15.9 | 15.9 |
| **% Rain minutes in autumn [%]** | 29.8 | 29.8 | 29.8 | 29.8 |
| **% Rain minutes 0.1–2 mm h$^{-1}$ [%]** | 78.1 | 77.2 | 72.4 | 73.8 |
| **% Rain minutes 2–5 mm h$^{-1}$ [%]** | 15.6 | 16.0 | 18.2 | 17.8 |
| **% Rain minutes 5–10 mm h$^{-1}$ [%]** | 4.6 | 4.9 | 6.7 | 6.1 |
| **% Rain minutes 10–25 mm h$^{-1}$ [%]** | 1.4 | 1.5 | 2.2 | 1.9 |
| **% Rain minutes > 25 mm h$^{-1}$ [%]** | 0.3 | 0.3 | 0.4 | 0.3 |
| **Highest rain rate [mm h$^{-1}$]** | 58.0 | 57.0 | 76.0 | 108.0 |

Figure 2 also shows the frequency distribution of the 318 rainfall events, which follow an exponential distribution with the majority of events lasting between 20 and 40 minutes; the exception is 4 events lasting far longer, with the longest covering
340 minutes. The total rainfall amounts recorded per event varied from 0.12 mm to 26.0 mm (with the instrument OTT3 taken as reference here). Over these three years, rainfall was frequent all year long, with more occurrences in the Southern Hemisphere autumn and winter, and summer accounting for the lowest rainfall occurrence. The mean average annual rainfall over these three years is within the mean envelope of rainfall records observed for this region over the last century.





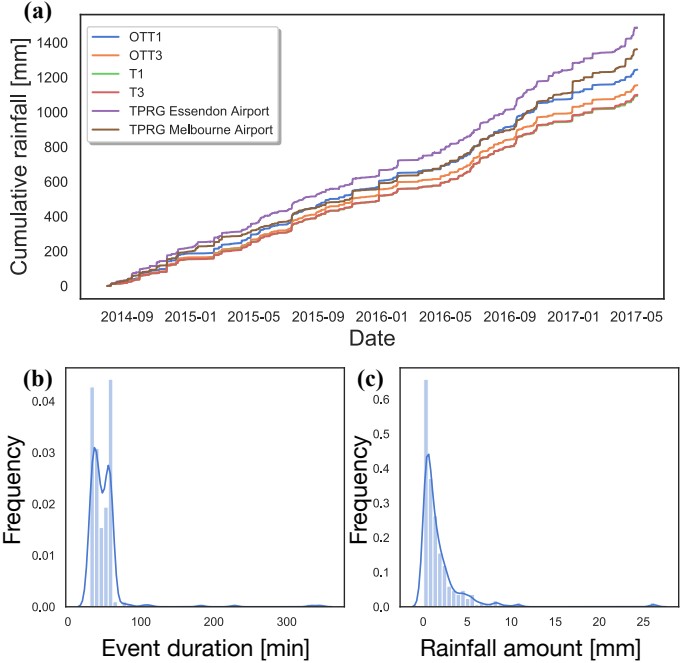


**Figure 2:** (a) Cumulative rainfall amount for the July 2014 to July 2017 period for the 4 disdrometers and two tipping bucket rain gauges located at 5.6 km (Essendon Airport) and 9.0 km (Melbourne airport); (b) Rainfall event duration frequency distribution; (c) Rainfall cumulative amounts per event frequency distribution.

**3.2 DSD for the full dataset**

After a succession of quality checks and filtering as described in the previous section, the total number of observed particles for each of the instruments was derived per bin size. In order to directly compare Thies LPM and OTT Parsivel[1], which have differences in resolution and measurement sensitivity (Table A1), common bin ranges were used and the results plotted in

Figure 3. Thies LPM instruments can measure smaller diameter drops and include a 0.125 to 0.25 mm bin size than OTT Parsivel[1], and therefore only Thies LPM observations are plotted for that diameter range. From the 0.25 to 0.5 mm range, the Thies LPM instruments measured a considerably larger number of droplets than OTT Parsivel[1], a feature documented in the literature (Chen et al., 2015; Angulo-Martinez et al., 2018) illustrating the higher sensitivity of the Thies LPM over the OTT Parsivel[1]. For the 0.5 to 0.75 mm range as well as the 1.5 to 3.0 mm range, a very consistent total number of droplets was

measured across all instruments of both manufacturers. OTT Parsivel[1] measured more droplets than the Thies LPM for the 0.75 to 1.5 mm range, which likely explains the higher total amount of rainfall measured over the three years by both OTT





Parsivel[1]. For larger drop diameter ranges (> 3 mm), the Thies LPM systematically recorded more droplets than the OTT Parsivel[1], but their total amount as compared to the middle range diameter bins were orders of magnitude less. Very small differences can be observed between instruments of the same manufacturer (*e.g.* T1 versus T3, or OTT1 versus OTT3), with

the larger discrepancies being observed at each end of the spectrum, *e.g.* for the lowest diameter bin for the Thies LPM, and the upper range of diameters for both OTT Parsivel[1] and Thies LPM.

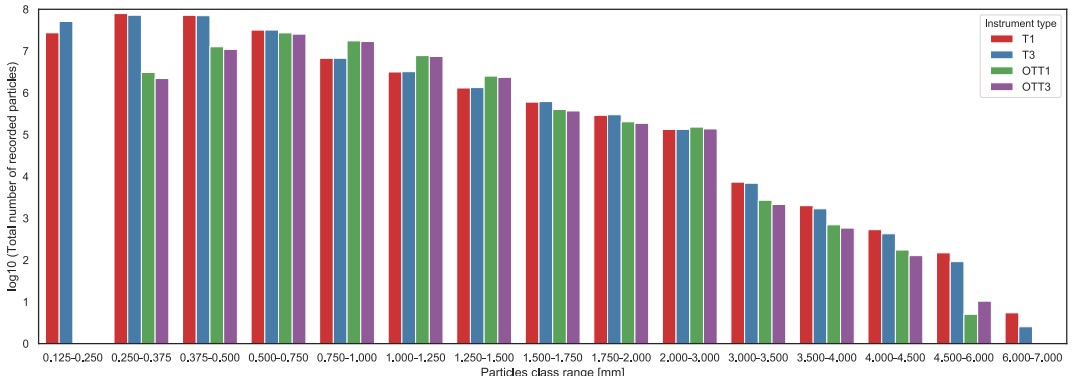

**Figure 3:** Distribution of the recorded total number of particles per bin for each of the four co-located instruments.


### 3.3 Detailed inter-comparison for a single event

The longest duration event in the data was event #7, which lasted for 344 minutes and produced more than 20 mm of rain. Figure 4 shows the time-series of integrated DSD variables (or moments) as well as DSD spectrum as density plots. To

facilitate readability, the full DSD graphs on Figure 4 (c) and (d) only show OTT1 and T1, as the instruments show very similar DSD characteristics across the event. For this event, the Thies LPM instruments presented a similar total rainfall amount ($\sum R$ = 19.7 and $\sum R$ = 20.3 mm for T1 and T3, respectively), lower than OTT1 and OTT3 ($\sum R$ = 22.8 mm and $\sum R$ =26.0 mm, respectively). Differences were small between instruments for low rainfall rates but large discrepancies are found for higher rainfall rates. OTT1 and OTT3 showed systematically higher rain rates than Thies LPM, corresponding to more

particles of large diameter recorded in the bins > 4 mm. There was also a larger number of droplets in the mid-range diameters. Conversely, the Thies LPM recorded more particles in total, as shown by the higher $N_w$ across the full event, with the largest differences for OTT Parsivel[1] at low rainfall rates. While these differences in total number of recorded particles are significant, they did not impact the horizontal reflectivity, which is sensitive to the number of large drops because $Z_H$ is equivalent to the 6[th] moment of the DSD. The OTT Parsivel[1] instruments still showed a slightly higher reflectivity for the

second part of high rain rates for this event (minutes 220 to 280).

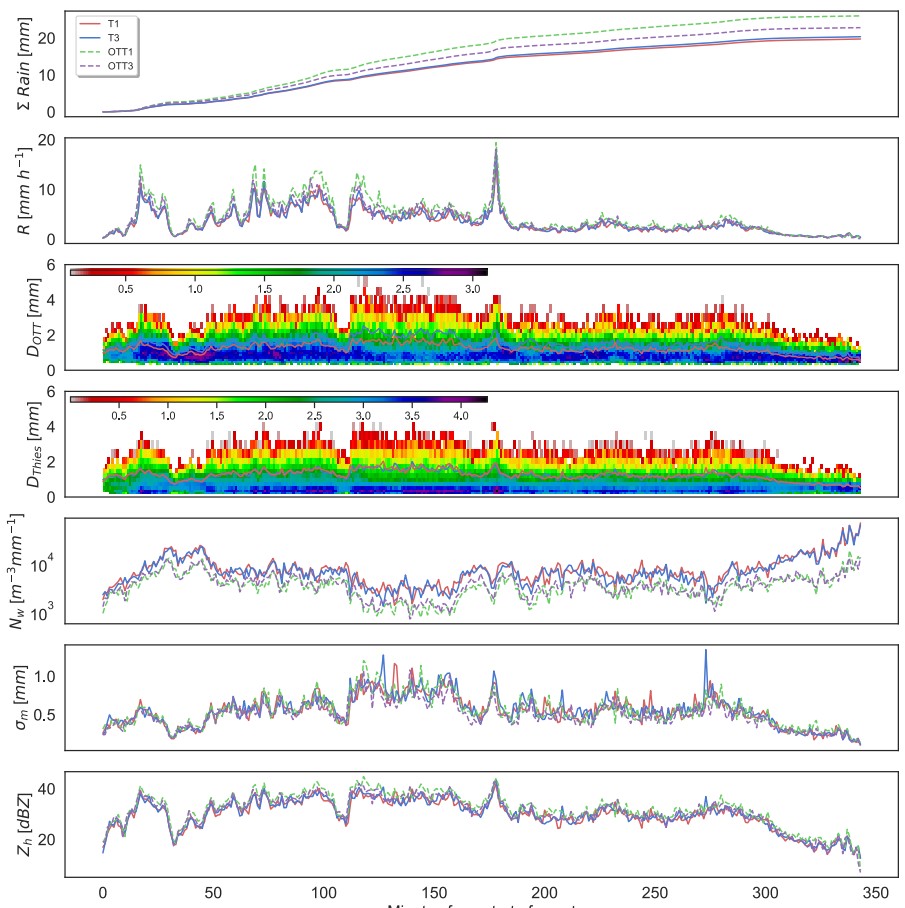

**Figure 4:** Rainfall Event #7 time-series of: (a) Cumulative rainfall amounts (mm); (b) Rainfall rate (mm h$^{-1}$); (c) Density distribution (log-scale) of drop size diameters for OTT1 (mm); (d) Density distribution (log-scale) of drop size diameters for T1 (mm); (e) Generalised intercept parameter (m$^{-3}$ mm$^{-1}$); (f) Mass spectrum standard deviation (mm); and (g) Horizontal reflectivity (dBZ).

### 3.4 Minute-resolution DSD

The Kernel Density Estimation (KDE) technique was used to estimate the DSD parameters for each of the 4062 minutes that had good quality data for all four instruments. The frequency distributions of these parameters are shown in Figure 5.



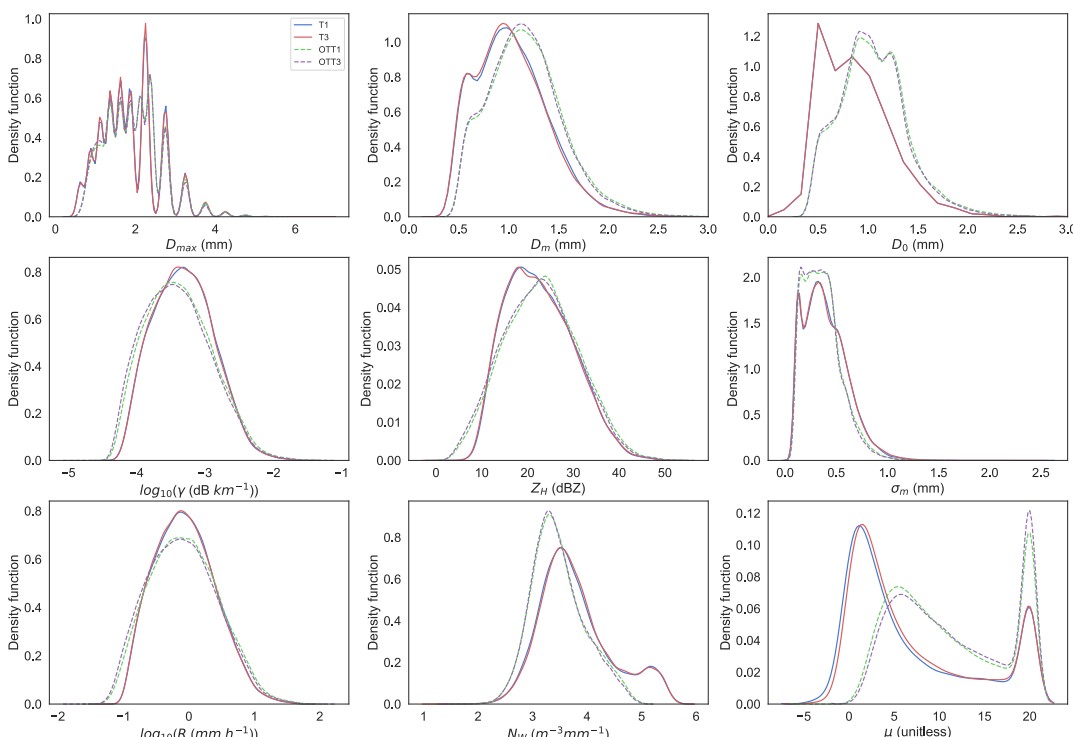

**Figure 5:** Density estimates of the DSD parameters ($D_{max}$, $D_m$, $D_0$, $\gamma_H$, $Z_H$, $\sigma_m$, $R$, $N_W$ and $\mu$) for the four instruments (T1, T3, OTT1, OTT3) for the full dataset.

The frequency distribution of the parameters exhibit very little differences from the same manufacturer (OTT or Thies), but large differences were found between Thies PLM and OTT Parsivel[1] for the lower order moments such as $D_m$, $D_0$, $\sigma_m$, $N_w$. A remarkable feature is the difference in the frequency distributions for $D_m$, $D_0$ and $\sigma_m$ due to the larger number of smaller droplets that were measured by the Thies LPM as compared to the OTT Parsivel[1]. These large numbers of particles recorded by the Thies LPM are identified in $N_w$ as the peak towards larger values, which is not found in the OTT Parsivel[1] statistics. The impact of this higher frequency of small droplets is a shift towards a smaller median reflectivity for the Thies LPM, while attenuation is on average higher for the Thies disdrometers as compared to the OTT Parsivel[1]. The shape of the





345    Gamma distribution $\mu_0$ utilised to parameterise the DSD is substantially different between OTT Parsivel[1] and Thies LPM due

to this larger number of smaller droplets for the Thies LPM. The $\mu$ distribution has a bimodal distribution with a first peak of

6 (OTT) or 0.5 (Thies) and a second peak at around 20 for both. This smaller value of $\mu_0$ for the Thies LPM corresponds to a

different shape of the DSD influenced by a larger number of small diameters.

**3.5 Effect of filtering and data correction on DSD parameters**

The effect of some of the filtering and the presence of smaller droplets on the DSD parameters is analysed here through the

means of the analysis shown in Figures 6 and 7. Figure 6 shows the PSVD matrix density plots for the full period of

observation for the four instruments, as well as the boundaries used to eliminate the erroneous data as proposed by Jaffrain

and Berne (2011), but using here the model of Atlas et al. (1973) instead of Beard (1977). The OTT Parsivel[1] sampled a

larger range of diameters and velocities as a result of their resolution and bin arrangement shown in Table A1. Relatively

few droplets fell into the zones of low velocity / large diameters or high velocity / small diameters, while one of Thies LPM

instruments (T3) measured a very large number of particles falling into these two categories. This was hypothesised to be

due to the design of the Thies LPM, increasing the probability to record more edge events (smaller sampling area) and the

brackets surrounding the laser beam increasing the possibility of splashes, both contributing to a higher number of falsely

identified rain droplets (Angulo-Martinez et al., 2018).

The effects of the filtering process on the frequency distribution of the DSD parameters can be seen in Figure 7. Only one

instrument of each manufacturer is shown for readability. The impact of the filtering was different for the two types of

instruments. Only the fitting parameters $N_w$ and $\mu_0$ were slightly affected for the Thies LPM. In contrast, the OTT Parsivel[1]

data were more affected, in particular for $R$, with more frequent smaller rainfall rates, due to the larger particles having been

removed, reducing the already smaller rainfall rates.

One can also quantify the contribution of small diameter ranges (< 0.6 mm) to the integrated variables through Figure 7. For

the Thies LPM, $D_m$ and $D_0$ showed a peak in the frequency distributions for the lower diameters above 0.6 mm after

removing the smaller bin size records, as expected. This is because their relative contribution increased when removing

highly frequent smaller bin ranges. This feature was not observed for the OTT Parsivel[1], showing that the small diameter

records were not much more frequent relative to the total amount of recorded particles. Also as expected, the proportion of

smaller rainfall rates was reduced for the Thies LPM when removing the smaller diameter bin sizes, since this instrument is

highly sensitive to smaller droplets that contribute to both small and high rain rates. DSD moments of higher order showed

less modification of their values when not considering diameter bins < 0.6 mm, since the smaller diameters contribute

relatively less when the moment order increases (order 6 for $Z_H$).


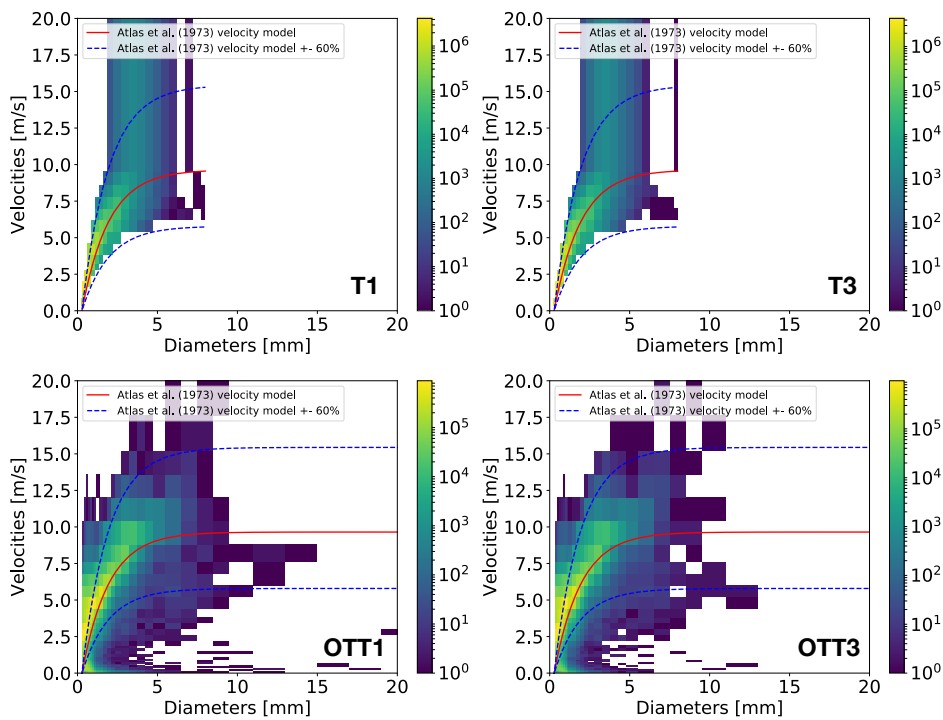

**Figure 6:** Particle velocity *versus* diameter density plots for the 2014-2017 period for the four instruments. The color scale indicates the number of particles for each bin (velocity and diameter). The velocity model of Atlas et al. (1973) is plotted in red together with its positive and negative deviations of 60%. This is used to filter outliers following Jaffrain and Berne (2011).





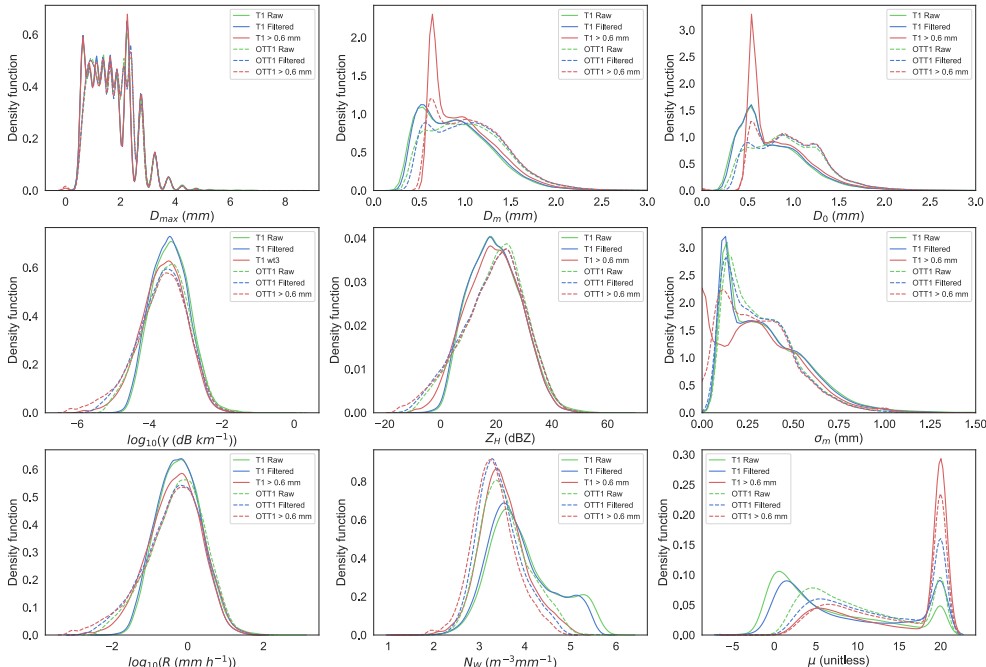


**Figure 7:** Frequency plots of the DSD parameters, based on distributions (not shown) of integrated variables ($D_{max}$, $D_m$, $D_0$, $\gamma_H$, $Z_H$, $\sigma_m$, $R$, $N_W$ and $\mu$) for two instruments (OTT1 and T1) for the full dataset for rainfall rates > 0.1 mm h$^{-1}$ and for raw data (green), filtered data (blue) and for minute data with mean diameter > 0.6 mm (red).

### 3.6 Properties of minute DSD variables for different rainfall rates


As seen in Event #7 (Figure 4), the DSD shape and integrated variables became increasingly different across the co-located instruments as the rainfall rate increased. In order to explore the effect of the rainfall rate, KDEs of the integrated variables from minute DSD data for the four co-located instruments are shown in Appendix A (Figures A1, A2, A3, A4) and in Figure

8 for rainfall rates of 0.1-2, 2-5, 5-10, 10-25 and > 25 mm h$^{-1}$, respectively. Figure 8 shows the most pronounced differences corresponding to the highest rain rates. Differences between instruments of the same manufacturer increased with rain rate, in particular for OTT Parsivel[1]. OTT1 showed more frequent higher rain rates, reflectivity and attenuation values than OTT3, while both Thies LPM and OTT3 statistics were similar. The first order moments started to show discrepancies between all instruments for rain rates > 10 mm h$^{-1}$, due to the sampling effect related to the occurrence of larger drops falling erratically

in space and time, therefore being captured by some instruments while not by co-located neighbours. Smaller values of $\mu_0$ for





the Thies LPM indicate the presence of a larger amount of small droplets (also seen in $N_W$), therefore modifying the shape of the DSD towards the smaller normalised diameters.

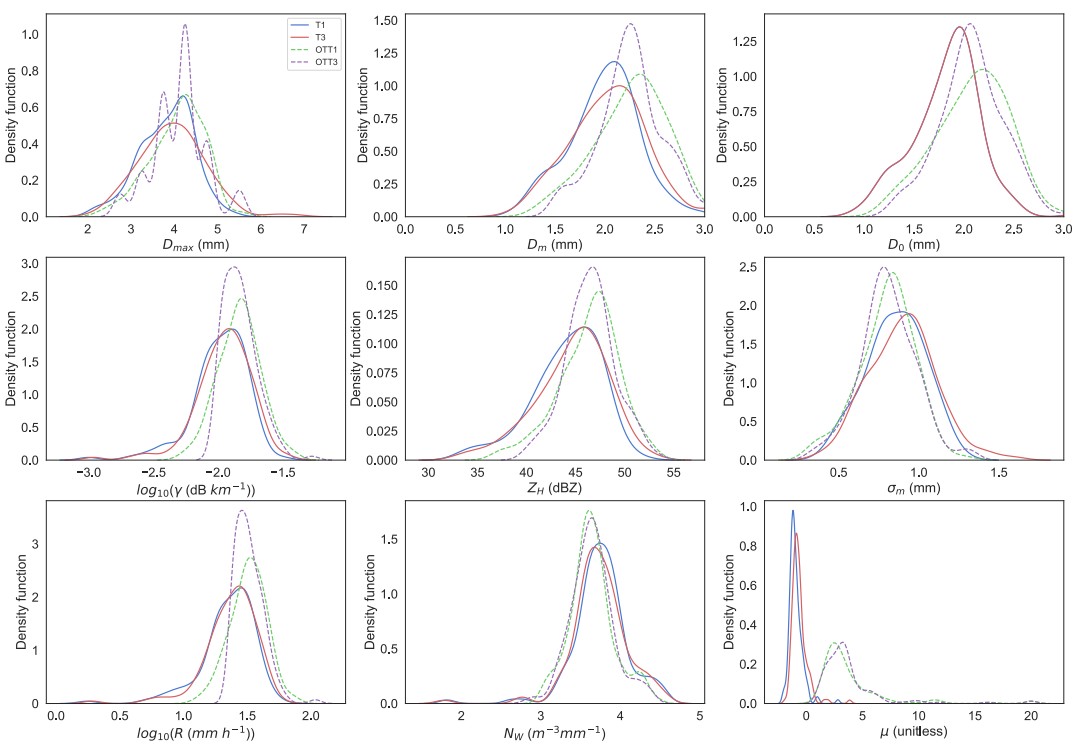


**Figure 8:** Frequency plots of the DSD parameters, based on distributions (not shown) of integrated variables ($D_{max}$, $D_m$, $D_0$, $\gamma_H$, $Z_H$, $\sigma_m$, $R$, $N_W$ and $\mu$) for the four instruments (T1, T3, OTT1, OTT3) for R > 25 mm h$^{-1}$ (129 data points).

**3.7 $Z_h$-R and $k$-R relations**

Horizontal reflectivity versus rainfall rate typically follows a simple power-law such as $Z_H = a\ R^b$, as initially described by Marshall and Palmer (1948), with the coefficients of that power law relation depending on the DSD properties (Uijlenhoet, 2001). Hence, it is of interest here to characterize the variability of such relationships among the four co-located instruments.





Figure 9 shows the scatter plots together with fitted $Z_H$-R relations, with the fitting being done for DSD $D_m < 0.6$ mm and $D_m > 0.6$ mm. Indeed, the occurrence of bimodal distributions as seen previously implies that if considering the full dataset, there should be at least two corresponding power-law relations for each distribution of the data. Distinct $Z_H$-R relations for two stratiform rainfall regimes (one corresponding to drizzle for $D_m <0.6$ mm and one corresponding to rain or rain and drizzle for $D_m > 0.6$ mm) as well as convective (for $Z_H > 30$ dBZ) are shown in Table 3.

The power law relations and related $a$ and $b$ coefficients were found to be very similar for the same instrument type (OTT Parsivel[1] or Thies LPM) but differed significantly between manufacturers when considering the full dataset for fitting the relations, and for convective rainfall. For stratiform rainfall ($D_m > 0.6$ mm) coefficients were almost identical for the four instruments, which was expected given the similarity in the observations of the DSD across all instruments for the middle range of diameter bins.

**Table 3:** $Z_h$-R relations for each of the co-located instruments corresponding to data fits to different subset of data.

| | OTT1 | OTT3 | T1 | T3 |
|---|---|---|---|---|
| **All data** | $Z_H = 253R^{1.47}$ | $Z_H = 256R^{1.47}$ | $Z_H = 220R^{1.42}$ | $Z_H = 220R^{1.41}$ |
| **Stratiform ($D_m > 0.6$ mm)** | $Z_H = 264R^{1.42}$ | $Z_H = 265R^{1.42}$ | $Z_H = 264R^{1.39}$ | $Z_H = 264R^{1.38}$ |
| **Stratiform ($D_m < 0.6$ mm)** | $Z_H = 55R^{1.09}$ | $Z_H = 55R^{1.08}$ | $Z_H = 50R^{1.01}$ | $Z_H = 50R^{1.01}$ |
| **Convective (for $Z_h > 30$ dBZ)** | $Z_H = 513R^{1.14}$ | $Z_H = 510R^{1.13}$ | $Z_H = 592R^{1.05}$ | $Z_H = 592R^{1.04}$ |

$Z_H$-R relations being directly related to the DSD, so are the coefficient $a$ and $b$. Empirical observations and further demonstrations (Uijlenhoet et al., 1999; 2001; 2003a) have shown that the value of coefficient $a$ will tend to increase and the value of coefficient $b$ to decrease as the raindrop mean diameter size increases and the concentration in the volume

decreases, e.g. in the case of convective rainfall, in particular for thunderstorms. This is indeed what is observed here for the relations derived for convective rainfall and in the range of values observed in the literature to date (Uijlenhoet et al., 2003b; 2006).

The $Z_H$-R relations for $D_m < 0.6$ mm correspond to drizzle, which consists of essentially small drops. The existing literature

on $Z_H$-R relations for drizzle is confined to a few articles with most of the observations having been made using aircraft-based instruments at cloud level. Comstock et al. (2004) report historical $Z_H$-R relations for drizzle with varying $a$ and $b$ coefficients: $Z_H = 150R^{1.5}$ (Joss et al., 1970) and $Z_H = 10R^{1.0}$ (Vali et al., 1998), with the Joss et al. (1970) $b$ coefficient being estimated and not measured. Comstock et al. (2004) also reported their own measurements of $Z_H = 25R^{1.3}$, $Z_H = 32R^{1.4}$ and $Z_H = 57R^{1.1}$. The estimates in Table 3 are within the range of those observations for drizzle, with a $b$ coefficient closer to unity

(indicating the so-called equilibrium precipitation, Uijlenhoet et al., 2003) while there was a reduced value of the $a$ coefficient as compared to rainfall. The vertical profile of DSDs above ground, in particular for stratiform clouds, is an entire





field of research, and further discussion and analysis on that aspect should be conducted in a dedicated study. The findings of this paper showed that the Thies LPM has the capacity to capture this part of the DSD spectrum.

**Table 4:** Differences in rainfall estimation following diverse fitted $Z_h$-R relations based each type (OTT or Thies) of instruments.

| 10 log ($Z_H$) [dBZ] | $Z_H$ [mm$^6$ mm$^{-3}$] | Difference in rain rate T1 *vs* OTT1 (all data) (in % and mm h$^{-1}$) | Difference in rain rate T1 *vs* OTT1 (convective) (in % and mm h$^{-1}$) | Difference in rain rate T1 *vs* OTT1 (drizzle) (in % and mm h$^{-1}$) |
|---|---|---|---|---|
| 0 | 1 | 4% and 0.001 mm h$^{-1}$ | - | 19% and 0.04 mm h$^{-1}$ |
| 10 | 10 | 0% and 0 mm h$^{-1}$ | - | 1% and 0.01 mm h$^{-1}$ |
| 20 | 100 | -5.7% and -0.03 mm h$^{-1}$ | - | - |
| 30 | 1000 | -10.8% and -0.27 mm h$^{-1}$ | 10.4% and 0.17 mm h$^{-1}$ | - |
| 40 | 10000 | -14.7% and -1.78 mm h$^{-1}$ | 4% and 0.6 mm h$^{-1}$ | - |
| 50 | 100000 | -20.4% and -11.8 mm h$^{-1}$ | 18.6% and 23.8 mm h$^{-1}$ | - |


Table 4 shows the difference in rainfall rates between different relations for a range of given reflectivity values, both in percentage and rainfall rate differences in mm h$^{-1}$. The differences across two instruments of the same manufacturer (Thies LPM versus Thies LPM or OTT versus OTT) were so minimal that these were not shown. Differences for stratiform rainfall ($D_m > 0.6$ mm) were not shown for the same reason. As can be seen in Table 4, the difference in rainfall rate estimates if

using OTT Parsivel[1] or Thies LPM DSD-derived $Z_H$-R relations fitted to all data shows disparities, increasing with higher values of reflectivity where differences reach -20.4% and- -11.8 mm h$^{-1}$. For smaller values of the reflectivity and up to 30 dBZ, OTT Parsivel[1] and Thies LPM DSD-derived $Z_H$-R relations gave reasonably equivalent estimates of the rainfall rates, with differences staying below 10%. For convective rainfall, the differences increased with increasing reflectivity and were equivalent to 23.8 mm h$^{-1}$ for 50 dBZ. For the stratiform drizzle, differences were large between instrument for small rainfall

but that is equivalent to small amounts. The main impact of the drizzle component was that these datapoints eventually change the $Z_H$-R relations when these are not excluded from the fitting for stratiform rainfall.



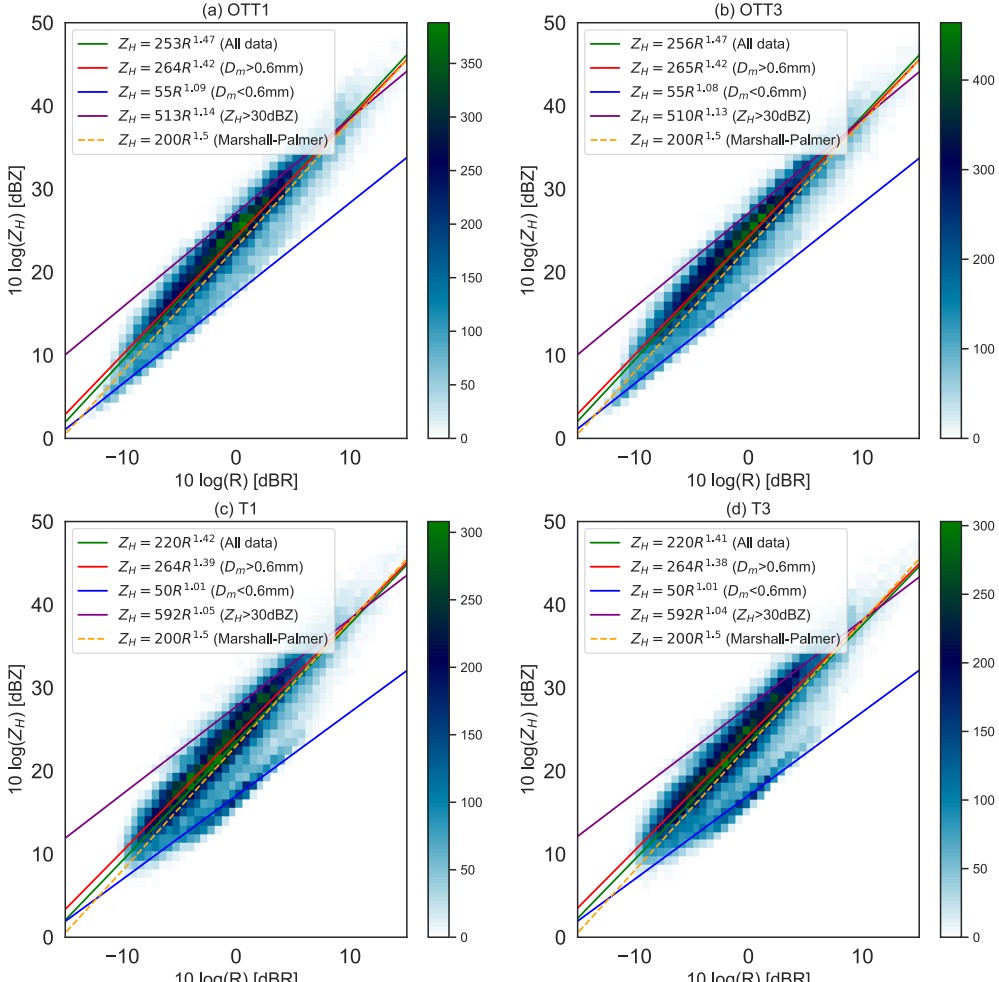

**Figure 9:** Horizontal reflectivity (in 10 log10 ($Z_H$) [dBZ]) versus rain rate (in 10 log10 (R) [dBR]) scatter plots for all co-located instruments (a to d). Power law relations are fitted separately for: all data, data meeting $D_m > 0.6$ mm (red curves), data meeting $D_m > 0.6$ mm (blue curves), data meeting $Z_H > 30$ dBZ (purple curves). The reference Marshall-Palmer relation is also shown.

Figure 10 shows the coefficients of the R-γ or γ-R relations derived from the DSD recorded by T1, both for horizontal and vertical polarisations. Correlation coefficients are shown both in their R-γ or γ-R forms, since the former is usually needed for estimating the impact of rain on CML to optimise the design of the network, while the latter is used to retrieve rainfall





from CML or other sources providing attenuation measurements. Most differences between ITU and the DSD-derived coefficients from this study occur in the 0-10 GHz range in the shape of the relation, and for the tail towards higher frequencies. These differences are large and lead to differences in rainfall intensities retrieved from values of γ of up to 30% if considering ITU or DSD-derived coefficients.


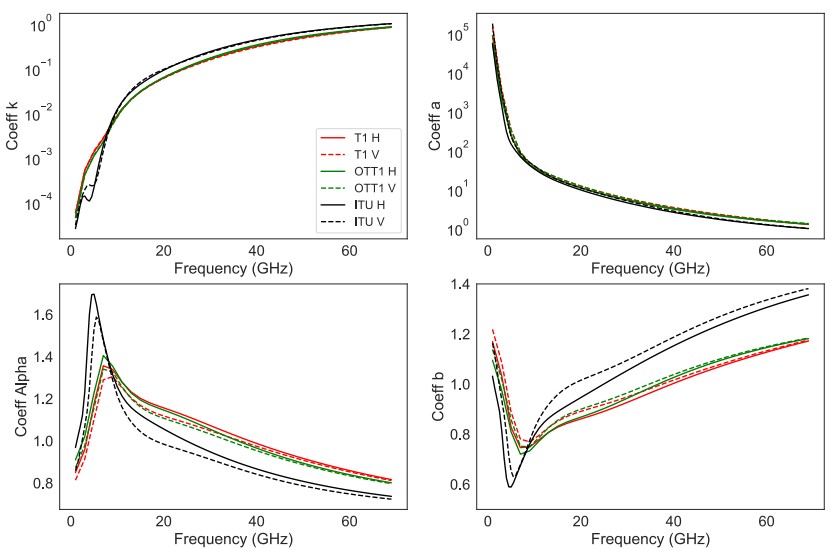

**Figure 10:** Coefficients of the γ-R or R-γ relations for both horizontal and vertical polarisations derived from the full DSD dataset for Thies LPM T1 and OTT Parsivel[1] OTT1. The black line represents the International Telecommunications Union model (ITU, 2005).

## 475    4. Discussion

Laser disdrometers are the most popular devices to observe and monitor the DSD, and from these the OTT Parsivel[1] and Parsivel[2], and the Thies LPM have been the most commonly used, both in the scientific literature to date (Fernandez-Raga et al., 2011) but also for operational applications by governmental and weather agencies. Despite this, few studies have
investigated the differences in raw and processed data as measured by these devices, with Angulo-Martinez et al. (2018) presenting for the first time a comparison between OTT Parsivel[2] and Thies LPM. Those authors gathered and analysed a dataset for a drier climate and for a shorter period than the one presented here, corresponding to approximately a quarter of the data and a fifth to a seventh of the rainfall accumulations over the study periods. A significant difference with Angulo-Martinez et al. (2018) is that they used the OTT Parsivel[2] while this study relied on an older version of this disdrometer, the
Parsivel[1]. Tokay et al., (2014) showed that the Parsivel[2] measured more droplets in the range 0.340 – 0.580 mm as compared





to Parsivel[1], while Parsivel[1] tends to absolutely over-estimate the number of larger drops over 2.40 mm size. Tokay et al., (2014) also showed that both Parsivel 1 and 2 generally measure consistent numbers of droplets in the medium 0.6 to 2.4 mm range. This can possibly explain the different results obtained by Angulo-Martinez et al. (2018) as compared to this work. This study confirmed some of their findings, namely (i) a systematic under-estimation of the total number of small

droplets by the OTT Parsivel as compared to the Thies LPM, when comparing identical bin ranges between 0.2 and 0.5 mm, leading to consistently smaller values of $D_m$ for the Thies LPM; (ii) more consistency between co-located Thies LPM than between co-located OTT Parsivel; and (iii) PSVD raw observations showing larger numbers of recorded non-raindrop (artefacts) size-velocity pairs for Thies LPM than for OTT Parsivel[1], likely due to more splashes and edge particles for the Thies LPM. However, the analysis of the present dataset showed opposite conclusions in terms of: (i) Total rainfall amounts,

with Thies LPM systematically recording lower total amounts than OTT; (ii) Integrated higher order moments: Reflectivity (and therefore attenuation, not presented in Angulo-Martinez et al. 2018) were smaller on average for Thies LPM as compared to OTT Parsivel[1]. This is due to a larger number of particles recorded by the OTT Parsivel as compared to the Thies LPM in the range 0.75 and 1.5 mm, diameters that contribute the most to the higher order moments because of the power law relation between diameters and these moments.


The potential sources and causes for errors and uncertainties that could explain observed differences between the instruments are discussed extensively in Angulo-Martinez et al. (2018), who also refer to previous work dedicated mostly to OTT Parsivel[1,2]. These authors list a number of possible causes including the geometry of the laser beams, internal software differences, design of the mounting structure and brackets, a known underestimation of falling velocity for OTT Parsivel

(Tokay et al., 2014). All of the above are likely contributing to cause most of the observed differences between the two types of instruments, but some of these issues will be challenging to solve, in particular in the absence of an independent absolute reference measurement of the DSD. High-frequency imagery (such as 2DVD disdrometers) has also been shown to have its own drawbacks and cannot be used as a reliable reference method, particularly for small diameters. Laser-based MPS has been hypothetically suggested as more reliable for measuring smaller drops (Thurai et al., 2017), but as other methods will

remain subject to biases and errors. Overall, co-located measurements using different instrument types, such as Thurai et al. (2017), Angulo-Martinez et al. (2018) and the present work are one of the possible ways of at least analysing the discrepancies between different measurement sources and discussing possible biases of each method, therefore providing a more robust estimate of the DSD spectra and derived integrated variables. In particular, such long-term high-resolution datasets are of importance for testing existing and new parameterisations of the DSD (Raupach et al., 2018; Thurai and

Bringi, 2018; Thurai et al., 2019), combine weakness and strengths of each sensor to derive more accurate DSD spectra or investigate the sampling and spatio-temporal characteristics of rainfall at micro-scale (Uijlenhoet et al., 2003ab, 2006; Jaffrain and Berne, 2011, 2012).





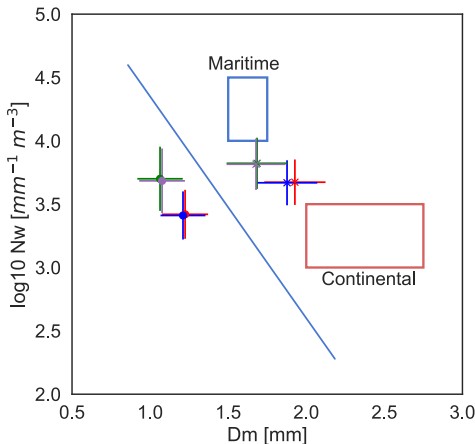

**Figure 11:** Average values of log10 $N_W$ versus $D_m$ for the data presented in this study (OTT1 in blue, OTT3 in red, T1 in green, T3 in purple) and their standard deviations. The blue line represents the demarcation between stratiform rain (dot points) (below the line) and convective rain (cross) (above the line), and the rectangle cover the clusters of data corresponding to maritime-like and continental-like as defined by Bringi et al., (2003) and augmented by numbers of authors since.

For the very first time, observations of the DSD have been presented for these temperate southern latitudes. In Australia, the only high-resolution DSD datasets have been obtained from experimental campaigns in the tropical north near Darwin (Maki et al., 2001; Giangrande et al., 2014) with only one short-term observation in subtropical Australia (Thurai et al., 2009). While the Southern Hemisphere in general lacks observations as compared to the Northern counterpart, Australia remains very poorly documented. Figure 11 shows the observations of the mean DSD variables from this current work on the climate regime chart as presented in Bringi et al., (2003), also showing the continental-like and maritime-like clusters the authors identified based on the existing DSD observations. These observations with the Thies LPM always yielded higher $N_W$ and $D_m$ as compared to the OTT Parsivel[1], reflecting the larger number of smaller droplets as observed by the Thies LPM. If considering the Thies LPM data more complete because of including a larger portion of the full DSD spectra, this brings the current observations closer to the maritime-like cluster identified by Bringi et al. (2003). This is in agreement with the hypothesis of a larger amount of small droplets generated in the precipitation originating from maritime environments.

**5. Conclusion**

This work presented for the very first time an open-access high-resolution PSVD dataset unique for this climate and location, including raw and carefully processed data with integrated DSD variables. This dataset could be further explored for a range

of studies, such as the environmental factors that can contribute to the observed DSD characteristics including diurnal precipitation type, seasonal and meso-scale precipitation variability, and the influence of oceanic or continental precipitation genesis. An obvious relevant future investigation given the main outcomes of this study would be the study of the effect of small diameter droplets (< 0.6 mm) on integrated DSD variables and on the parameterisation of the DSD. As the present work has shown the capacity of the Thies LPM to capture this part of the drop size spectrum with high sensitivity, a study

could build on this dataset and the recent findings from Thurai and Bringi (2018), Thurai et al. (2019) and Raupach et al. (2018) to study the parameterisation of the DSD, including the small droplets and the implication for retrievals such as Williams et al. (2014). Here, a new DSD dataset is provided for these latitudes and climatic region, together with reflectivity - rainfall rate and attenuation - rainfall rate relationships, relevant for rainfall parameterisation and retrievals for ground- and satellite-based radars as well as microwave links. Lastly, similarly to other studies presenting co-located observations, this

dataset gives the opportunity to study sampling effects and spatio-temporal characteristics of rainfall microphysics.

**Software and model codes**

pyDSD and pyTmatrix are openly available through GitHub repositories.

**Data availability**

The raw PSVD dataset presented in this study are publically available at http://www.doi:10.5281/zenodo.3234218

**Author contribution**

AG and JP developed the python pipeline to process the data based mainly on pyDSD and pyTmatrix libraries. AG and JP performed the data analysis. All co-authors provided ideas and feedback following discussions. AG prepared the manuscript with contributions from all co-authors.

**Acknowledgments**

The authors would like to thank the technical staff from the Australian Bureau of Meteorology (BoM) for the installation and maintenance of the disdrometers, in particular David Wright and Tom Kane. An Australian Research Council Discovery Project Grant number DP160101377 funded this project. The authors declare no conflict of interest.

**Appendix A**





**Table A1:** Observation ranges of the Thies LPM and OTT Parsivel[1] disdrometers for diameter and velocity bins.

| Bin diameter range (mm) Thies LPM | Bin diameter range (mm) OTT Parsivel[1] | Bin velocity range (m s⁻¹) Thies LPM | Bin velocity range (m s⁻¹) OTT Parsivel[1] |
|---|---|---|---|
|  | *0.000 to 0.125* | 0.0 to 0.2 | 0.0 to 0.1 |
| 0.125 to 0.250 | *0.125 to 0.250* |  | 0.1 to 0.2 |
| 0.250 to 0.375 | 0.250 to 0.375 | 0.2 to 0.4 | 0.2 to 0.3 |
|  |  |  | 0.3 to 0.4 |
|  | 0.500 to 0.625 | 0.4 to 0.6 | 0.4 to 0.5 |
| 0.500 to 0.750 |  |  | 0.5 to 0.6 |
|  | 0.625 to 0.750 | 0.6 to 0.8 | 0.6 to 0.7 |
|  |  |  | 0.7 to 0.8 |
| 0.750 to 1.000 | 0.750 to 0.875 | 0.8 to 1.0 | 0.8 to 0.9 |
|  | 0.875 to 1.000 | 1.0 to 1.4 | 0.9 to 1.0 |
| 1.000 to 1.250 | 1.000 to 1.125 | 1.4 to 1.8 | 1.0 to 1.2 |
|  | 1.125 to 1.250 | 1.8 to 2.2 | 1.2 to 1.4 |
| 1.250 to 1.500 | 1.250 to 1.500 | 2.2 to 2.6 | 1.4 to 1.6 |
| 1.500 to 1.750 | 1.500 to 1.750 | 2.6 to 3.0 | 1.6 to 1.8 |
| 1.750 to 2.000 | 1.750 to 2.000 | 3.0 to 3.4 | 1.8 to 2.0 |
| 2.000 to 2.500 | 2.000 to 2.250 | 3.4 to 4.2 | 2.0 to 2.4 |
| 2.500 to 3.000 | 2.250 to 2.575 | 4.2 to 5.0 | 2.4 to 2.8 |
|  | 2.575 to 3.000 | 5.0 to 5.8 | 2.8 to 3.2 |
| 3.000 to 3.500 | 3.000 to 3.500 | 5.8 to 6.6 | 3.2 to 3.6 |
| 3.500 to 4.000 | 3.500 to 4.000 | 6.6 to 7.4 | 3.6 to 4.0 |
| 4.000 to 4.500 | 4.000 to 4.500 | 7.4 to 8.2 | 4.0 to 4.8 |
| 4.500 to 5.000 |  | 8.2 to 9.0 | 4.8 to 5.6 |
| 5.000 to 5.500 | 4.500 to 5.125 | 9.0 to 10.0 | 5.6 to 6.4 |
| 5.500 to 6.000 | 5.125 to 6.000 | > 10.0 | 6.4 to 7.2 |
| 6.000 to 6.500 |  |  | 2.2 to 8.0 |
| 6.500 to 7.000 | 6.000 to 7.000 |  | 8.0 to 9.6 |
| 7.000 to 7.500 |  |  | 9.6 to 11.2 |
| 7.500 to 8.000 | 7.000 to 8.000 |  | 11.2 to 12.8 |
|  | 8.000 to 9.000 |  | 12.8 to 14.4 |
|  | 9.000 to 10.250 |  | 14.4 to 16.0 |
|  | 10.250 to 12.000 |  | 16.0 to 19.2 |
|  | 12.000 to 14.000 |  | 19.2 to 22.4 |
|  | 14.000 to 16.000 |  |  |
| > 8.000 | 16.000 to 18.000 |  |  |
|  | 18.000 to 20.000 |  |  |
|  | 20.000 to 23.000 |  |  |
|  | 23.000 to 26.000 |  |  |

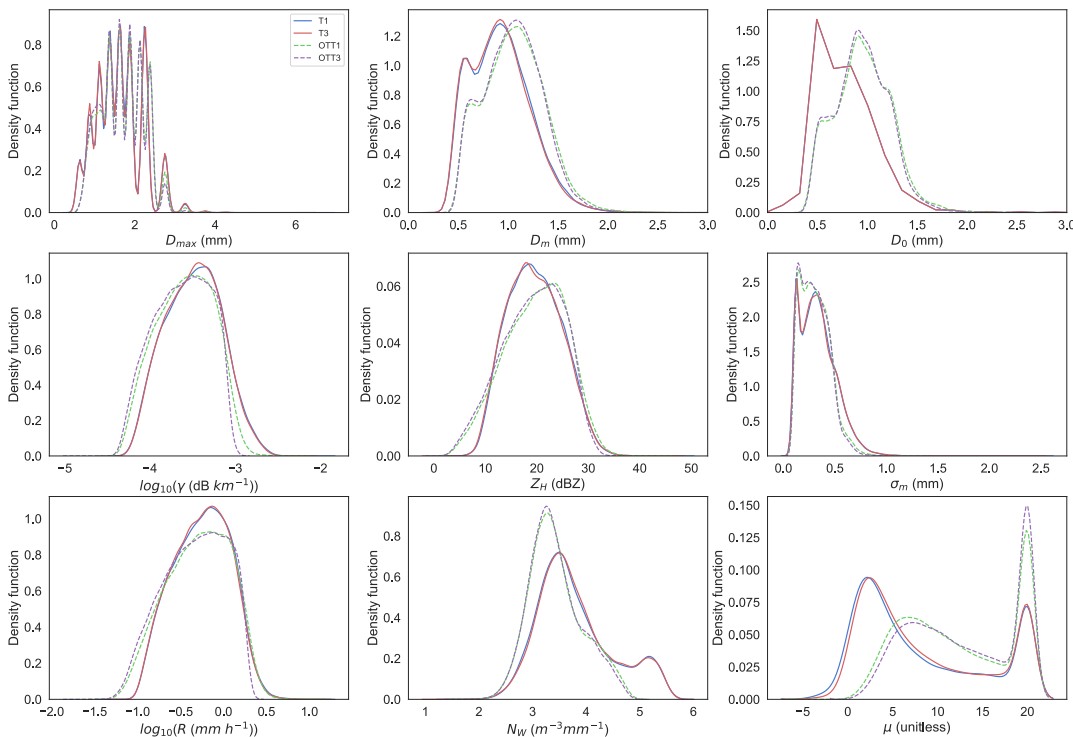


**Figure A1.** Frequency plots of the DSD parameters, based on distributions (not shown) of integrated variables ($D_{max}$, $D_m$, $D_0$, $\gamma_H$, $Z_H$, $\sigma_m$, $R$, $N_W$ and $\mu$) for the four instruments (T1, T3, OTT1, OTT3) for rainfall rates of 0.1 to 2 mm h$^{-1}$ (29815 data points).



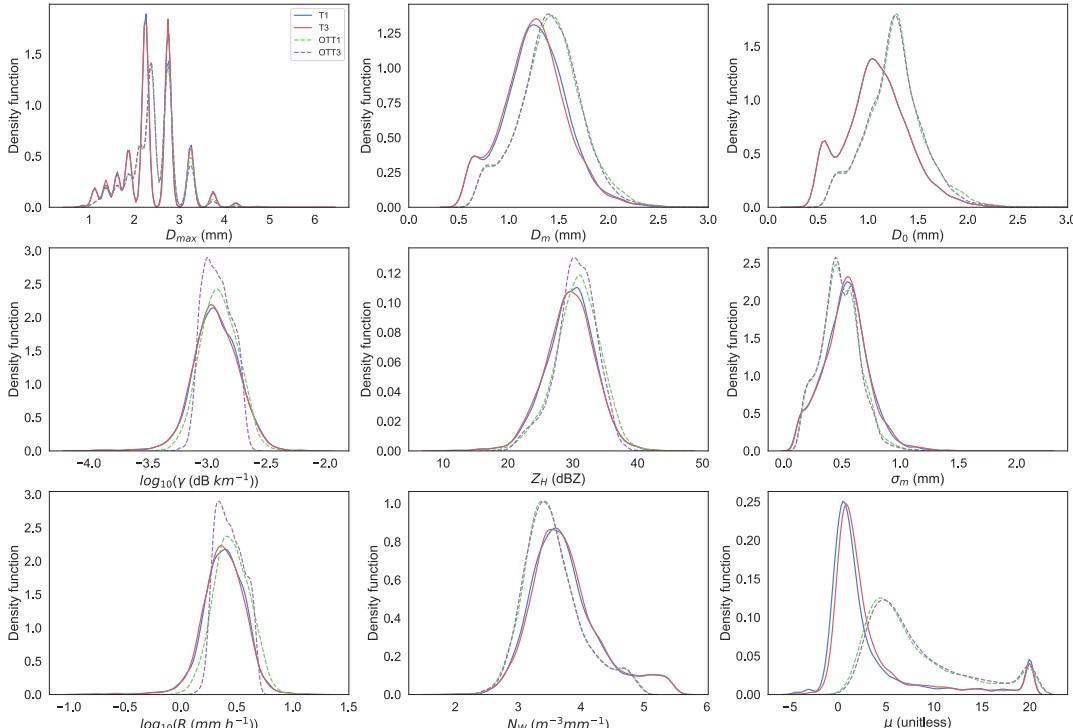

**Figure A2.** Frequency plots of the DSD parameters, based on distributions (not shown) of integrated variables ($D_{max}$, $D_m$, $D_0$, $\gamma_H$, $Z_H$, $\sigma_m$, $R$, $N_W$ and $\mu$) for the four instruments (T1, T3, OTT1, OTT3) for rainfall rates of 2 to 5 mm h$^{-1}$ (6967 data points).



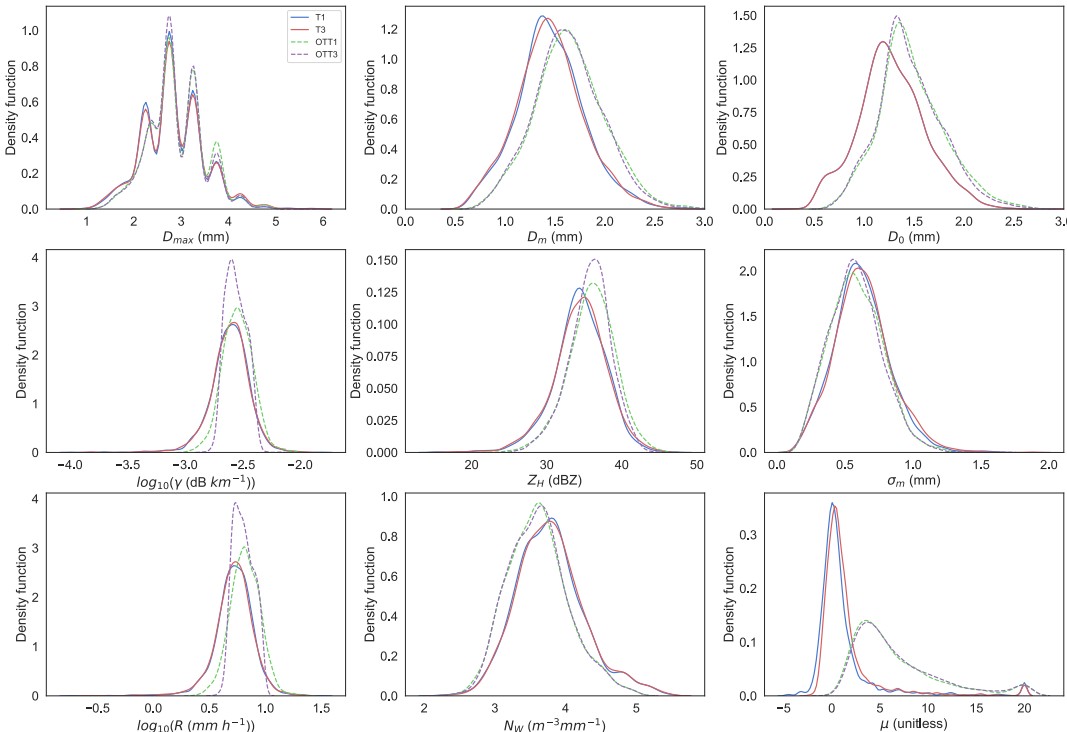


**Figure A3.** Frequency plots of the DSD parameters, based on distributions (not shown) of integrated variables ($D_{max}$, $D_m$, $D_0$, $\gamma_H$, $Z_H$, $\sigma_m$, $R$, $N_W$ and $\mu$) for the four instruments (T1, T3, OTT1, OTT3) for rainfall rates of 5 to 10 mm h$^{-1}$ (2398 data points).



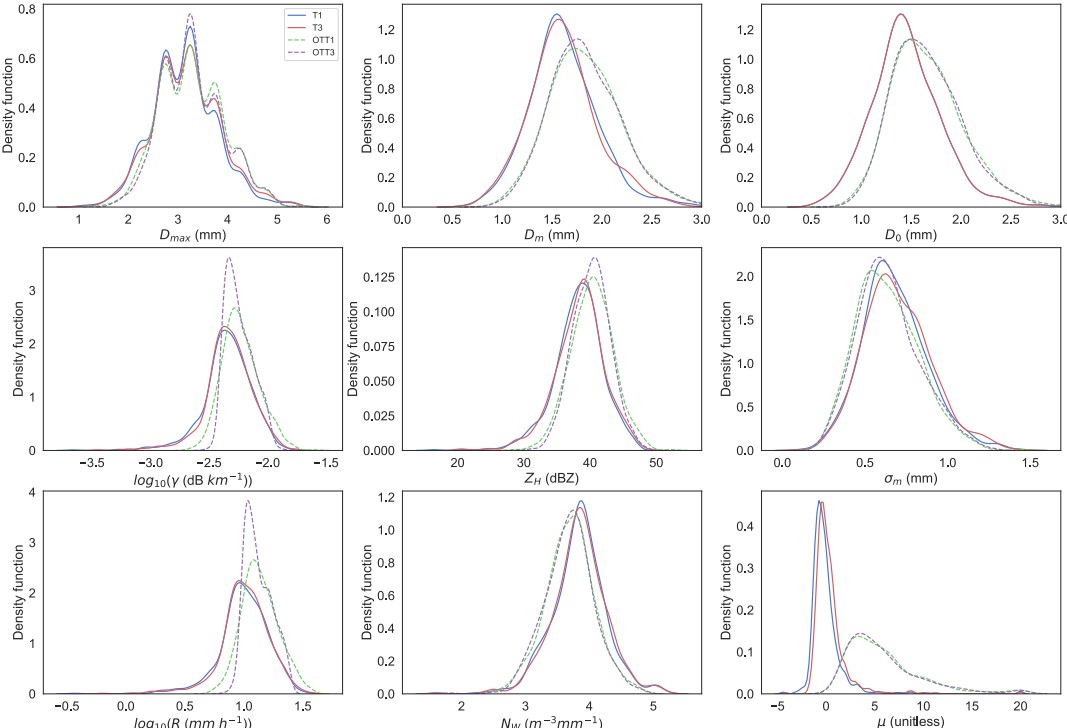

**Figure A4.** Frequency plots of the DSD parameters, based on distributions (not shown) of integrated variables ($D_{max}$, $D_m$, $D_0$, $\gamma_H$, $Z_H$, $\sigma_m$, $R$, $N_W$ and $\mu$) for the four instruments (T1, T3, OTT1, OTT3) for rainfall rates of 10 to 25 mm h$^{-1}$ (753 data points).

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
