# Peer review of "Effect of disdrometer type on rain drop size distribution"

_Hydrology and Earth System Sciences, 2019_

## Referee Comment (RC1) · Anonymous Referee #1 · 10 Jul 2019

Review of "Effect of disdrometer type on rain drop size distribution characterisation: a new dataset for Southeastern Australia" by Guyot et al.

The article presents a detailed comparison of drop size distribution (DSD) measurements taken by four collocated instruments (by two different manufacturers) located in Australia. Such southern-hemisphere comparison studies are uncommon, especially for the mid-latitudes. The study is clearly organised, well written and presents a thorough analysis. The results are useful and future directions are outlined. Some minor changes are required before the article will be ready for publication: there are occasional grammar errors and spelling mistakes that should be fixed in the next revision.

At times more references should be provided (I have indicated below when this is the case). In a few cases, the statements made in the text were not supported by the figures, and these require clarification. It is significant that the DSD database collected by the authors is freely available for use.

Specific comments follow:

1. Lines 70–72: This section is rather light on references; please include some of the pioneering studies about microstructure and its effect on QPE. I would think scattering properties, being instantaneous, depend more on microstructure than microphysics as such.

2. Line 73: The DSD describes microstructure, not microphysics (unless changes in the DSD are studied over time).

3. Line 76: References should be provided for stain and oil immersion techniques.

4. Line 87: The reference Thurai et al. 2017 seems to be missing from the references list.

5. Line 109: A reference should be provided for the 20 years of observations near Darwin.

6. Line 140: Please also specify how far apart the two instrument types were and their relative orientation.

7. Line 160: $N_t$ is usually defined as the particle or drop concentration and given in m$^{-3}$. Is this $N_t$ different? It is also listed as unitless in Table 1.

8. Line 189: Should "min" be "minute"?

9. Table 1: Please double-check the units for PSVD (decibel seems an odd choice) and the symbol used for rainfall amount (amount is not a summation of hourly rain rate).

10. Line 201: The range around the expected velocity should be 60% to match the reference and Figure 6.

11. Line 214: Please show how $A_i$ was calculated; since depending on the Parsivel version used, the formula used to calculate $A_i$ differs (whether or not $D$ or $D/2$ is used should depend on whether or not the Parsivel automatically removes drops detected in the edge region).

12. Line 217 and Eq. 2: This version of $Z$ is in mm$^6$ m$^{-3}$, not dBZ.

13. Line 224: Is this canting angle the standard deviation of an angle distribution?

14. Line 228: "attentuations" → "specific attenuations".

15. Line 238: $\Lambda$ has unit of mm$^{-1}$. Which fitting method was used to find the ordinary gamma model parameters?

16. Line 266: By my reading the Parsivels showed more than 100 mm difference in cumulated amount.

17. Line 268: This is first mention of a second tipping bucket 9 km away; it should be introduced alongside the first gauge in Section 2.5.

18. Table 2: What does "Equivalent" mean in this instance? I assume the cumulated amounts are for all rainy minutes per instrument, not over the 40062 common time steps?

19. Line 290: "can measure smaller diameter drops and include a 0.125 to 0.25 mm bin size than OTT" – sentence doesn't make sense.

20. Line 300: What explanation is there for the differences being greatest at the ends of the spectrum? I would guess sensitivity differences for the Thies differences for small drops, and sampling effects for the large drops.

21. Figure 3: why do no Parsivels record any drops larger than 6 mm? (Were they removed, or were there simply no recorded drops by the OTTs?). This is also strange given that in the example event in the next section, OTTs record larger numbers of larger drops.

22. Line 308: It would be useful to include the time (or at least month/season) of the event.

23. Line 329: Please include a reference for KDE. KDE is used to estimate probability distributions of observed variables, not to estimate the DSD parameters for each minute as written here.

24. Figure 4: What are the lines in the density distribution plots? Please also label the plots a) to g) to match the caption.

25. Figure 5: $D_{max}$ and $D_0$ should be defined in the text. The "spiky" density estimated for $D_{max}$ is presumably due to the discrete diameter classes used and would disappear if different KDE bandwidth were used. Incidently, is $D_{max}$ here calculated on the shared classes? If not I would expect the densities to differ by instrument just because of the different class definitions.

26. Line 345: $\mu_0$ and $\mu$ are used interchangeably here. It would be of interest to show the mean DSD per instrument (ie mean/median and bars for quantiles on $N(D)$ by $D$ class) to empirically show the differences in shape.

27. Lines 357–358: "one of the Thies LPM instruments (T3) measured a very large number of particles falling into these two categories" – I do not see this very large

number of particles in Figure 6, which shows more particles outside the expected velocity ranges for OTT1 and OTT3.

28. Figure 7: Please clarify "mean diameter" as meaning DSDs with small drops removed, since "mean diameter" could be taken to mean $D_m$. The discussion on lines 369–373 is confusing and requires rephrasing – is the point that the Thies instruments seem to have a lot of drops in the first class after 0.6 mm, where as the drops are more evenly distributed in the OTT cases?

29. Lines 374–375 and Figure 7: I interpret the plot for $R$ in Figure 7 as showing that without the small drops there are more very low rain rates, since for example the left tail on the solid red line is left of the blue and green lines; this appears to clash with the statements made in the article text. In the OTT distributions and the T1 $> 0.6$ distribution there are rain rate values less than 0.1 mm h$^{-1}$, which was stated to be the minimum allowed in these analyses. What explains these low values?

30. Lines 391–392: The statement here (that differences between instruments are larger as $R$ increases) is not supported by Figure 4, in which there is not a clear effect on the differences that correlates with the peaks in $R$.

31. Lines 397–398: The statements here are not supported by the plots in Figure 8. OTT1 shows similar frequencies to OTT3 for for high reflectivity, attenuation, and $R$. The big difference is that in the OTT1 distributions there are more low values and less frequent mid-range values than in OTT3 distributions.

32. Line 398: Which variables are meant by "first order" moments here? Since high rain rates mean many drops of all sizes, the stated link between high rain rates and large-drop sampling uncertainty requires some more argument, e.g. by looking specifically at the variables influenced more by large drop occurrences ($D_m$, $Z$). I think that sampling uncertainty due to sample numbers decreasing with

increasing rain rates may play a much larger role than the large drops in the observed differences (e.g. there are only 129 points in Figure 8, but 29815 in Figure A1).

33. Lines 415–419: It is not clear why it is important to separate $D_m$ at 0.6 mm, or which previous bimodal distributions the authors are referencing. A reference to the scheme used to separate convective from stratiform regimes should be included here.

34. Lines 428–430: The statement that $b$ decreases with increasing $D_m$ is not true when comparing the fits on the two stratiform data sets, and because all data contains the convective data it is hard to compare the results for "all" to the convective results.

35. Line 444: While true that this paper showed Thies can capture small drops related to drizzle, no estimate of the measurements' accuracy can be made without another instrument that also captures those drop sizes.

36. Table 4: $\log \rightarrow \log_{10}$, and $\text{mm}^6\,\text{mm}^{-3} \rightarrow \text{mm}^6\,\text{m}^{-3}$. Over what range of $Z_H$ values were these differences calculated; i.e. for each value in the left-most column, what was the class size in dBZ? Which instrument was taken as the reference, i.e. the percentage is of which value of $R$?

37. Figure 9: The logarithms should be specified as $\log_{10}$ in the axis labels. Some brief discussion in the main text about the differences between the fitted relationships and the normal Marshall-Palmer $Z$-$R$ relation should be included. Also, the $b$ exponent in the Marshall-Palmer version is 1.6 not 1.5 as stated in the plot key; please include a reference to Marshall 1955 in the caption for this plot and clarify which relationship is shown in the plot.

38. Figure 11 caption: "augmented by numbers of authors since" – which authors? Please provide references.

39. Line 544: Another possible future direction could be comparisons of Thies LPM to other instruments that, unlike Parsivel, are able to accurately measure concentrations of small drops.

40. Line 545: The reference to Raupach et al. 2019 has year 2019 in the references list but 2018 in the text.

41. Lines 525–535: The appearance and discussion of Figure 11 do not fit well into this paragraph, which is ostensibly about a lack of DSD observations in Australia. Are the authors aiming to highlight the mismatch between their observations and the climate regimes shown in Figure 11? This discussion feels incomplete.

---

## Referee Comment (RC2) · Anonymous Referee #2 · 26 Jul 2019

Guyot et al. describes a new 3 year dataset for southeastern Australia collected from two different manufacturers of optical disdrometers. The authors have prepared a careful analysis of the differences for instruments from the same manufacturer and instruments from different manufacturers. Significant differences were documented due to the sampling sensitivity at different droplet sizes and velocities that results in changes to the derived DSD. The paper's treatment of the scientific objectives is robust and no significant issues could be found. Recommend accepted with technical corrections.

Technical corrections: Page 4 Line 18: Reference needed for Darwin observations

Page 5 Lines 10-13 contains too many ideas - needs to be broken up into two sen-

tences Line 12: Understanding the synoptic rainfall regimes is important for such a study. Has any previous work been done that you can reference? Figure 1(b): Melbourne map is difficult to read - can you increase the contrast and maybe add a border? Figure 1 caption: Are the stands at 2m or 1.5m (as stated in the previous caption)

Page 10 Equation(2): Units of Zmom should be Z instead of dBZ?

Page 11 Line 6: Do you mean specific attenuation? This needs to be clarified.

Page 7 General comment: It sounds like the optical disdrometers are using a laser beam sheet with negligible depth to sample the DSD? Maybe worth stating this explicitly for readers.

Page 9 Line 5: Drizzle/Rain repeated in brackets

Page 11 Line 3: How dependent is the T-Matrix calculations on the temperature? It seems a 20C temperature might bias towards warmer rainfall events? Line 23: Two rain gauges are referred two, but only one is introduced.

Page 12 Line 11: the sentence starting with 'The recorded...' needs more context. maybe say 'This erroneous data...' Table 2: What does 'high quality refer to? Figure 2 (b)(c) caption: Are the duration/intensity analysis derived from rain gauges or disdrometers? Figure 3: Why are there no OTT stats for the 6-7mm class? Figure 4: subplot labels are missing and description of lines in the density distribution plots

Page 18 Line 17: It's not clear to me where T3 exhibits significantly more size/velocity samples outside the outliers in figure 6. It looks like T1 has more outliers, and the OTT's even more so.

Page 20 Line 14: What does the author mean by 'first order moments'?

data availability section Page 28: the url should not include the 'www' subdomain, just http://doi.org/10.5281/zenodo.3234218 given this paper promotes the underlying data as 'open source' or 'open access', it would be ideal to include some description of

exactly what data has been hosted on zenodo (which I can't check because it's under embargo). e.g., what instruments are provided and what the file format it.

---

## Referee Comment (RC3) · Anonymous Referee #3 · 9 Aug 2019

The paper present a comprehensive analyses of drop size distribution (DSD) measurements using 2 pairs of laser-based disdrometer instruments from two manufacturers (OTT and Thies Clima) installed on the same observational site. The measurements took place in Melbourne, Australia, between 2014 and 2017. Raw and the processed disdrometer data were analysed with the objective to evaluate their differences, provide a quantitative description of the DSD for the region and the local climate, as well as develop relationships for horizontal reflectivity – rainfall rate (Zh-R) and attenuation – rainfall rate (gamma-R) of the microwave radiation. The paper is well written and organized, it meets the objectives with well-developed discussion. It concludes with directions for future investigations and provides the research community with an open

access to the raw DSD dataset. The manuscript could be accepted after the technical corrections.

Comments:

Line 87: the reference Thurai et al. (2017) is missing in the reference list.

Line 95: incorrect reference according to the reference list – de Moraes Frasson et al., 2011.

Line 105: first two references are missing in the reference list.

Line 140 and Figure 1: not clear whether the disdrometers are placed 1.5 or 2 m above ground.

Line 169 (and elsewhere): Appendix 1 is in conflict with the title on page 27 – Appendix A.

Line 201: the reference Atlas et al. (1973) is missing in the reference list.

Line 219: the reference Testud et al. (2001) is missing in the reference list.

Line 237: the reference Ulbrich (1983) is missing in the reference list.

Line 294: the reference Chen et al. (2015) does not have an exact match in the reference list (2016).

Line 339: mistype – Thies PLM.

Lines 428, 431 and 440: the references Uijlenhoet et al. (2003a, 2003b and 2003) are not well defined according to the reference list.

Line 437: the reference Joss et al. (1973) is missing in the reference list.

Line 461: incorrect description of the blue curves (Dm < 0.6 mm).

Line 478: the reference Fernandez-Raga et al. (2011) is missing in the reference list.

[Figure]

Line 508: the used acronym MPS has not been previously described.

Line 514: the reference Raupach et al. (2018) does not have an exact match in the reference list (2019).

Lines 516 and 517: the reference Uijlenhoet et al. (2003ab) should be corrected, Jaffrain and Berne (2011 and 2012) do not have a perfect match in the reference list.

Line 524: the reference Bringi et al. (2003) is missing in the reference list.

Line 594 – References: the references in the following lines have not been used in the paper: 628, 662, 666, 693, 702, 729 781 and 785.
* * *

---

## Author Comment (AC1) · 5 Sep 2019

**Responses to reviewer #3**
*Guyot et al.* **under review at HESS**

**https://doi.org/10.5194/hess-2019- 277, 2019**

We would like to thank the three reviewers for their very constructive comments on our manuscript. We received genuine insights, which have significantly contributed to increasing the manuscript quality and potential impact.

In order to improve the clarity in our responses we have numbered the reviewers' comments for reviewer #2 and #3 (Reviewer #1's comments are already numbered): for example, the comment 1 from reviewer 2 is listed as R1C2 and will refer to these comments as such in the following.

We have addressed all comments in point-by-point responses.

**REVIEWER #3**

The paper present a comprehensive analyses of drop size distribution (DSD) measurements using 2 pairs of laser-based disdrometer instruments from two manufacturers (OTT and Thies Clima) installed on the same observational site. The measurements took place in Melbourne, Australia, between 2014 and 2017. Raw and the processed disdrometer data were analysed with the objective to evaluate their differences, provide a quantitative description of the DSD for the region and the local climate, as well as develop relationships for horizontal reflectivity – rainfall rate (Zh-R) and attenuation – rainfall rate (gamma-R) of the microwave radiation. The paper is well written and organized, it meets the objectives with well-developed discussion. It concludes with directions for future investigations and provides the research community with an open access to the raw DSD dataset. The manuscript could be accepted after the technical corrections.

We would like to thank the reviewer for her/his time and insightful comments that will help improve the manuscript. The referencing has been done without Endnote or BibteX, which explains the numerous mismatches. Lesson learnt!

Comments:

R3C1: Line 87: the reference Thurai et al. (2017) is missing in the reference list.

Response: Reference added as:

Thurai, M., K.V. Mishra, V.N. Bringi, and W.F. Krajewski: Initial Results of a New Composite-Weighted Algorithm for Dual-Polarized X-Band Rainfall Estimation. *Journal of Hydrometeorology,* **18**, 1081–1100, https://doi.org/10.1175/JHM-D-16-0196.1, 2017.

R3C2: Line 95: incorrect reference according to the reference list – de Moraes Frasson et al., 2011.

Response: Thanks, now corrected.

R3C3: Line 105: first two references are missing in the reference list.

Response: First reference is added and second reference is deleted as it is a conference paper (first reference is enough).

R3C4: Line 140 and Figure 1: not clear whether the disdrometers are placed 1.5 or 2 m above ground.

Response: Disdrometers were installed at 1.5 m above ground level. This has been corrected in the Figure caption.

R3C5: Line 169 (and elsewhere): Appendix 1 is in conflict with the title on page 27 – Appendix A.

Response: Appendix 1 in the text has been changed to "Appendix A".

R3C6: Line 201: the reference Atlas et al. (1973) is missing in the reference list.

Response: Reference has been added to the reference list.

R3C7: Line 219: the reference Testud et al. (2001) is missing in the reference list.

Response: Reference has been added to the reference list.

R3C8: Line 237: the reference Ulbrich (1983) is missing in the reference list.

Response: Reference has been added to the reference list.

R3C9: Line 294: the reference Chen et al. (2015) does not have an exact match in the refer- ence list (2016).

Response: This has been corrected.

R3C10: Line 339: mistype – Thies PLM.

Response: This has been corrected.

R3C11: Lines 428, 431 and 440: the references Uijlenhoet et al. (2003a, 2003b and 2003) are not well defined according to the reference list.

Response: This has been fixed.

R3C12: Line 437: the reference Joss et al. (1973) is missing in the reference list.

Response: Joss and Waldvogel has been added to the reference list:

Joss, J. and A. Waldvogel: Raindrop Size Distribution and Sampling Size Errors. *J. Atmos. Sci.,* **26**, 566–569, https://doi.org/10.1175/1520 0469(1969)026<0566:RSDASS>2.0.CO;2, 1969.

And changed in the text to Joss and Waldvogel (1969).

R3C13: Line 461: incorrect description of the blue curves (Dm < 0.6 mm).

Response: The Figure caption has been corrected accordingly.

R3C14: Line 478: the reference Fernandez-Raga et al. (2011) is missing in the reference list.

Response: This reference is now removed as it was a discussion paper in AMT but was rejected, so it is unsuitable for citation.

R3C15: Line 508: the used acronym MPS has not been previously described.

Response: This is now described in the introduction (Page 3 line 90):

"Thurai et al. (2017) presented data from a Meteorological Particle Spectrometer (MPS) (Baumgardner et al., 2002), arguing its higher sensitivity and better accuracy for diameters below 1.1 mm as compared to the 2DVD, while the 2DVD was proven to be accurate above 0.7 mm."

R3C16: Line 514: the reference Raupach et al. (2018) does not have an exact match in the reference list (2019).

Response: Reference has been changed to Raupach et al. (2019) in the text.

R3C17: Lines 516 and 517: the reference Uijlenhoet et al. (2003ab) should be corrected, Jaffrain and Berne (2011 and 2012) do not have a perfect match in the reference list.

Response: This has been corrected.

R3C18: Line 524: the reference Bringi et al. (2003) is missing in the reference list.

Response: Reference has been added to the reference list.

R3C19: Line 594 – References: the references in the following lines have not been used in the paper: 628, 662, 666, 693, 702, 729 781 and 785.

Response: All of these references have been deleted except Verdon-Kidd and Kiem (2009), which has been kept and referenced in the text of the manuscript.

New appendix Figures as per below:

[Figure]

Revised Figure A1 above

[Figure]

Revised Figure A2 above

[Figure]

Revised Figure A3 above

[Figure]

Revised Figure A4 above

---

## Author Comment (AC2) · 5 Sep 2019

**Responses to reviewer #2**
*Guyot et al.* **under review at HESS**

**https://doi.org/10.5194/hess-2019- 277, 2019**

We would like to thank the three reviewers for their very constructive comments on our manuscript. We received genuine insights, which have significantly contributed to increasing the manuscript quality and potential impact.

In order to improve the clarity in our responses we have numbered the reviewers' comments for reviewer #2 and #3 (Reviewer #1's comments are already numbered): for example, the comment 1 from reviewer 2 is listed as R1C2 and will refer to these comments as such in the following.

We have addressed all comments in point-by-point responses.

**REVIEWER #2**

Guyot et al. describes a new 3 year dataset for southeastern Australia collected from two different manufacturers of optical disdrometers. The authors have prepared a careful analysis of the differences for instruments from the same manufacturer and instruments from different manufacturers. Significant differences were documented due to the sampling sensitivity at different droplet sizes and velocities that results in changes to the derived DSD. The paper's treatment of the scientific objectives is robust and no significant issues could be found. Recommend accepted with technical corrections.

We would like to thank the reviewer for his/her time helping us improve the quality of the manuscript. Some of her/his comments are consistent with the comments from reviewer #1, and when that happened, we flagged these with cross-referencing (such as R1C21 for R2C14) for the response.

R2C1: Technical corrections: Page 4 Line 18: Reference needed for Darwin observations

Response: Two references have been added (Dolan et al., 2013; Thomason et al., 2018). This was also a comment from R1 (R1C5).

R2C2: Page 5 Lines 10-13 contains too many ideas - needs to be broken up into two sentences

Response: Done.

R2C3: Line 12: Understanding the synoptic rainfall regimes is important for such a study. Has any previous work been done that you can reference?

Response: Thanks; we referenced this in an earlier version of the manuscript but somehow the reference got deleted in the text (but still remained in the reference list). We have now added again the same reference here (Verdon-Kidd and Kiem, 2009).

R2C4: Figure 1(b): Melbourne map is difficult to read - can you increase the contrast and maybe add a border? Figure 1 caption: Are the stands at 2m or 1.5m (as stated in the previous caption)

Response: Disdrometers were installed at 1.5 m above ground level. This has been corrected in the Figure caption. We added a border to the Melbourne map and augmented its size and contrast. Below is our new Figure 1:

[Figure]

R2C5: Page 10 Equation(2): Units of Zmom should be Z instead of dBZ?

Response: units are $mm^6\ m^{-3}$. This has been corrected.

R2C6: Page 11 Line 6: Do you mean specific attenuation? This needs to be clarified.

Response: This has now been clarified following also the same remark from reviewer #1 (see R1C14).

R2C7: Page 7 General comment: It sounds like the optical disdrometers are using a laser beam sheet with negligible depth to sample the DSD? Maybe worth stating this explicitly for readers.

Response: A sentence has been added to reflect that technical assumption.

R2C8: Page 9 Line 5: Drizzle/Rain repeated in brackets

Response: Duplication has been deleted.

R2C9: Page 11 Line 3: How dependent is the T-Matrix calculations on the temperature? It seems a 20C temperature might bias towards warmer rainfall events?

Response: Sensitivity of the T-matrix calculation to temperature was tested following a similar approach to Louf et al. (2019), but we decided not to expand on this as it goes beyond the scope of the current work. The retrievals are not very sensitive to the temperature but more to the canting angle and choice of model. This is not critical for single polarisation but impacts more the dual pol moments. We added the following sentence to our manuscript: "Sensitivity analysis of the T-matrix to the canting angle

and temperature, as well as a consistency analysis following a similar approach as Louf et al. (2019), were tested but not presented herein as this is beyond the scope of the present study."

R2C10: Line 23: Two rain gauges are referred two, but only one is introduced.

Response: The other rain gauge is now introduced as well. It reads:

"Another gauge located at Melbourne Airport (Bureau of Meteorology station #086282) and situated 9.0 km from the experimental site was also used for comparison."

R2C11: Page 12 Line 11: the sentence starting with 'The recorded...' needs more context. maybe say 'This erroneous data...'

Response: It is actually the non-erroneous data! We have now specified accordingly and added a reference to Table 2 to improve clarity. The new sentence reads:

"This corresponded to a total of 40,062 common quality ("quality" being defined as filtered and quality-checked data following the processing steps as described in the method section) minutes across the four instruments [...]"

R2C12: Table 2: What does 'high quality refer to?

Response: We have now clarified this terminology in two locations:

In the method section: "The post-processed data following these sequential steps is further described as "quality" data."

In the results section: "This corresponded to a total of 40,062 common quality ("quality" being defined as filtered and quality-checked data following the processing steps as described in the method section) minutes across the four instruments, with cumulative rainfall ranging from 1093 to 1244 mm (depending on sensor) over the observational period."

R2C13: Figure 2 (b)(c) caption: Are the duration/intensity analysis derived from rain gauges or disdrometers?

Response: This data is from OTT1. We have now changed the legend and it reads:

"**Figure 2:** (a) Cumulative rainfall amount for the July 2014 to July 2017 period for the 4 disdrometers and two tipping bucket rain gauges located at 5.6 km (Essendon Airport) and 9.0 km (Melbourne airport); (b) Rainfall event duration frequency distribution based on rainfall records from OTT1; (c) Rainfall cumulative amounts per event frequency distribution based on rainfall records from OTT1."

R2C14: Figure 3: Why are there no OTT stats for the 6-7mm class?

Response: Thanks; See also R1C21 who raised the same issue.

In order to process the data to plot Figure 3, we had to find overlapping bin classes for OTT and THIES instruments, which is shown in Table A1 (in appendix). We realised that there was an overlapping class category missing in the Table A1 (e.g. 0.375 to 0.500 mm

range). The corresponding pipeline python code therefore had the same error and that led to a shift in the corresponding bin class for the OTTs. Correcting this, the new Figure shows more particles counted for OTT1 and OTT3. Looking into the details, only 8 minutes for OTT1 and 3 minutes for OTT3 for the full dataset present particles falling into that bin class (6 to 7 mm). In Figure 6 the log scale magnifies the importance of that data.

R2C15: Figure 4: subplot labels are missing and description of lines in the density distribution plots

Response: See also R1C24. The red and blue lines in the density plot represent respectively the mean diameter $D_m$ for OTT1 and OTT3 (panel c) and T1 and T3 (panel d). The caption of the figure has been updated. Labels (a) to (g) have been added to the plot.

R2C16: Page 18 Line 17: It's not clear to me where T3 exhibits significantly more size/velocity samples outside the outliers in figure 6. It looks like T1 has more outliers, and the OTT's even more so.

Response: See also R1C27. Figure 6 is now updated showing also the non-filtered particles (outside of the boundaries defined using Atlas et al. (1973) model of fall velocity. Our apologies for that mistake: the OTT figure were the correct version showing non-filtered and filtered particles while the Thies LPM were showing only the filtered particles. Figure 6 now supports this statement as you can see for T3 in particular.

R2C17: Page 20 Line 14: What does the author mean by 'first order moments'?

Response: This is now replaced by "All DSD parameters (Figure 8)". See also R1C32.

R2C18: data availability section Page 28: the url should not include the 'www' subdomain, just http://doi.org/10.5281/zenodo.3234218 given this paper promotes the underlying data as 'open source' or 'open access', it would be ideal to include some description of exactly what data has been hosted on zenodo (which I can't check because it's under embargo). e.g., what instruments are provided and what the file format it.

Response: We have now added information to this section and it reads:

"The dataset presented in this study are publically available at http://doi:10.5281/zenodo.3234218. This includes raw data for each of the four disdrometers (OTT1, OTT3, T1 and T3) recorded as daily "telegrams" by the in-built software of each instrument. Fields include the proprietary software-derived integrated variables and PSVD data."
* * *
New appendix Figures as per below:

[Figure]

Revised Figure A1 above

[Figure]

Revised Figure A2 above

[Figure]

Revised Figure A3 above

[Figure]

Revised Figure A4 above

---

## Author Comment (AC3) · 5 Sep 2019

We would like to thank the three reviewers for their very constructive comments on our manuscript. We received genuine insights, which have significantly contributed to increasing the manuscript quality and potential impact.

In order to improve the clarity in our responses we have numbered the reviewers' comments for reviewer #2 and #3 (Reviewer #1's comments are already numbered): for example, the comment 1 from reviewer 2 is listed as R1C2 and will refer to these comments as such in the following.

We have addressed all comments in point-by-point responses.

**REVIEWER #1**

Review of "Effect of disdrometer type on rain drop size distribution characterisation: a new dataset for Southeastern Australia" by Guyot et al.
The article presents a detailed comparison of drop size distribution (DSD) measurements taken by four collocated instruments (by two different manufacturers) located in Australia. Such southern-hemisphere comparison studies are uncommon, especially for the mid-latitudes. The study is clearly organised, well written and presents a thorough analysis. The results are useful and future directions are outlined. Some minor changes are required before the article will be ready for publication: there are occasional grammar errors and spelling mistakes that should be fixed in the next revision.
At times more references should be provided (I have indicated below when this is the case). In a few cases, the statements made in the text were not supported by the figures, and these require clarification. It is significant that the DSD database collected by the authors is freely available for use.

Response: We would like to thank the reviewer for his/her insightful and detailed comments below. These helped greatly to improve the quality of the manuscript. We greatly appreciate that the reviewer has taken the time to provide such high quality comments. We provide a point-by-point response to the comments below:

Specific comments follow:
1. Lines 70–72: This section is rather light on references; please include some of the pioneering studies about microstructure and its effect on QPE. I would think scattering properties, being instantaneous, depend more on microstructure than microphysics as such.

Response: We added to the three references already cited in our paper (as per below) by including two new ones (Uijlenhoet and Sempere Torres, 2006; Krajewski and Smith, 2002). We have changed "microphysics" to "microstructure".
Uijlenhoet, R., J.A. Smith, and M. Steiner: The microphysical structure of extreme precipitation as inferred from ground-based raindrop spectra. Journal of Atmospheric Sciences, 60, 1220–1238, doi: 10.1175/1520-0469(2003)60<1220%3ATMSOEP>2.0.CO%3B2, 2003.

Uijlenhoet, R., M. Steiner, and J.A. Smith: Variability of raindrop size distributions in a squall line and implications for radar rainfall estimation. Journal of Hydrometeorology, 4, 43–61, doi:10.1175/1525-7541(2003)004<0043:VORSDI>2.0.CO;2, 2003.

Uijlenhoet, R.: Raindrop size distributions and radar reflectivity–rain rate relationships for radar hydrology, Hydrology and Earth System Sciences, 5(4), 615-628, https://doi.org/10.5194/hess-5-615-2001, 2001.

Uijlenhoet, R., Sempere Torres, D., Measurement and parameterization of rainfall microstructure, Journal of Hydrology, Volume 328, Issues 1–2, 2006, Pages 1-7, ISSN 0022-1694, https://doi.org/10.1016/j.jhydrol.2005.11.038.

W.F. Krajewski, J.A. Smith, Radar hydrology: rainfall estimation, Advances in Water Resources, Volume 25, Issues 8–12, 2002, Pages 1387-1394, ISSN 0309-1708, https://doi.org/10.1016/S0309-1708(02)00062-3.

2. Line 73: The DSD describes microstructure, not microphysics (unless changes in the DSD are studied over time).

Response: We have changed "microphysics" to "microstructure".

3. Line 76: References should be provided for stain and oil immersion techniques.

Response: We added the following references:

Fuchs, N., & Petrjanoff, I. (1937). Microscopic examination of fog-, cloud-and rain droplets. *Nature*, *139*(3507), 111.

Nawaby, A. S. (1970). A method of direct measurement of spray droplets in an oil bath. *Journal of agricultural engineering research*, *15*(2), 182-4.

Kathiravelu, G., Lucke, T., & Nichols, P. (2016). Rain drop measurement techniques: a review. *Water*, *8*(1), 29.

4. Line 87: The reference Thurai et al. 2017 seems to be missing from the references list.

Response: The reference has been added to the reference list.

5. Line 109: A reference should be provided for the 20 years of observations near Darwin.

Response: Two references have been added (Dolan et al., 2013; Thomason et al., 2018).

6. Line 140: Please also specify how far apart the two instrument types were and their relative orientation.

Response: We have now changed the sentence and it reads:

"A distance of 2 meters separated the Thies Clima LPM (T3) and the OTT Parsivel1 (OTT1) located on the edges of each of the rails (as seen in Figure 1). The laser

beams of each sensor were oriented along the North-South axis with raw 1-min data collected."

7. Line 160: usually defined as the particle or drop concentration and given in m$^{-3}$. Is this different? It is also listed as unitless in Table 1.

Response: According to the OTT Parsivel manual, it is specified that Nt is unitless (or min$^{-1}$ as this is a number per time step, with time-steps equal to minutes in our case) and calculated by the instrument internal software, based on the PSVD.

8. Line 189: Should "min" be "minute"?

Response: This has been corrected.

9. Table 1: Please double-check the units for PSVD (decibel seems an odd choice) and the symbol used for rainfall amount (amount is not a summation of hourly rain rate).

Response: This has been corrected changed to "unitless".

10. Line 201: The range around the expected velocity should be 60% to match the reference and Figure 6.

Response: Thanks: Indeed, this was a typo and has now been corrected.

11. Line 214: Please show how calculated; since depending on the Parsivel version used, the formula used to calculate (whether or not D or D/2 is used should depend on whether or not the Parsivel automatically removes drops detected in the edge region).

Response: The equation for removing edge droplets has been added as equation (2) and subsequent equations have been re-allocated an appropriate numbering.

12. Line 217 and Eq. 2: This version of Z is in mm$^6$ m$^{-3}$, not dBZ.

Response: This is now corrected.

13. Line 224: Is this canting angle the standard deviation of an angle distribution?

Response: Yes, it is probabilistic. We conducted some research (unpublished work) on the sensitivity of the canting angle on the T matrix retrievals following the same approach as Louf et al. (2019) by comparing ground based DSD and radar observed dual pol moments, using the self-consistency technique. These results are beyond the scope of the work presented here so have not been included.

14. Line 228: "attentuations" → "specific attenuations".

Response: This has been changed.

15. Line 238: Λ has unit of mm$^{-1}$. Which fitting method was used to find the ordinary gamma model parameters?

Response: Thanks, we added units to the text. The "Moments method" of Ulbrich and Atlas (1998) has been used to derive the parameters. A sentence has been added to the end of the paragraph and the reference of Ulbrich and Atlas (1998) to the reference list.

16. Line 266: By my reading the Parsivels showed more than 100 mm difference in cumulated amount.

Response: The exact reading based on Table 2 is 89 mm (derived from the absolute value of (1244 mm – 1155 mm)) but it exceeds 100 mm when comparing Parsivel to LPM. We have modified the text as per below:

"The two Thies LPM systems recorded very similar rainfall totals, while the two OTT Parsivel[1] systems showed a difference of 89 mm between them and above 100 mm when compared to the Thies LPM during the common observational period."

17. Line 268: This is first mention of a second tipping bucket 9 km away; it should be introduced alongside the first gauge in Section 2.5.

Response: Thanks. The sentence below was added to section 2.5:

"Another gauge located at Melbourne Airport (Bureau of Meteorology station #086282) and situated 9.0 km from the experimental site was also used for comparison."

18. Table 2: What does "Equivalent" mean in this instance? I assume the cumulated amounts are for all rainy minutes per instrument, not over the 40062 common time steps?

Response: No: these are for the 40,062 common time steps. We have added this to the column label in order to clarify.

19. Line 290: "can measure smaller diameter drops and include a 0.125 to 0.25 mm bin size than OTT" – sentence doesn't make sense.

Response: Indeed, that was rather obscure… we have corrected and now it reads:

"The Thies LPM instruments can measure smaller diameter drops as they include a 0.125 to 0.25 mm bin size which the OTT Parsivel[1] does not cover. Therefore, only Thies LPM observations are plotted for that diameter range."

20. Line 300: What explanation is there for the differences being greatest at the ends of the spectrum? I would guess sensitivity differences for the Thies differences for small drops, and sampling effects for the large drops.

Response: Yes that is the most reasonable hypothesis. With the error (see comment below), this is even clearer for larger drops for the OTT instruments. We added a sentence such as:

"The observed differences for the lowest diameter bins are likely due to sensitivity differences between the Thies LPM instruments, while the differences for the largest diameter particle class range (6 to 7 mm) seen across all four instruments are likely

due to sampling effects. The observed particles for the range 6 to 7 mm correspond to only 3 or 8 recorded minutes (out of 40,062) depending on the instrument."

21. Figure 3: why do no Parsivels record any drops larger than 6 mm? (Were they removed, or were there simply no recorded drops by the OTTs?). This is also strange given that in the example event in the next section, OTTs record larger numbers of larger drops.

Response: In order to process the data to plot Figure 3, we had to find overlapping bin classes for the OTT and THIES instruments, as shown in Table A1 (in appendix). We realised that there was an overlapping class category missing in the Table A1 (e.g. 0.375 to 0.500 mm range). The corresponding pipeline python code therefore had the same error and that led to a shift in the corresponding bin class for the OTTs. Correcting this, the new Figure shows more particles counted for OTT1 and OTT3. Looking into the details, only 8 minutes for OTT1 and 3 minutes for OTT3 for the full dataset present particles falling into that bin class (6 to 7 mm). In Figure 6 the log scale magnifies the importance of that data.

Below is our new Figure 3:

[Figure]

Below is our new Figure 6:

[Figure]

22. Line 308: It would be useful to include the time (or at least month/season) of the event.

Response: The event happened on the 24th September 2014 starting in the morning at 9:09am local time (AEDT). We have now included that information in the Figure caption.

23. Line 329: Please include a reference for KDE. KDE is used to estimate probability distributions of observed variables, not to estimate the DSD parameters for each minute as written here.

Response: Thanks, a reference has been added and the text modified accordingly.

24. Figure 4: What are the lines in the density distribution plots? Please also label the plots a) to g) to match the caption.

Response: The red and blue lines in the density plot represent respectively the mean diameter $D_m$ for OTT1 and OTT3 (panel c) and T1 and T3 (panel d). The caption of the figure has been updated and labels (a) to (g) have been added to the plot. Here below is the new Figure 4:

[Figure]

25. Figure 5: Dmax should be defined in the text. The "spiky" density estimated for D max is presumably due to the discrete diameter classes used and would disappear if different KDE bandwidths were used. Incidentally, is Dmax here calculated on the shared classes? If not I would expect the densities to differ by instrument just because of the different class definitions.

Response: The definition of $D_{max}$ has been added to the text next to the existing definition of $D_m$. We have tested different bandwidths for the KDE and modified the figures using the optimised bandwidth for each of the figures presenting KDEs (Figures 5, 7 and 8, and appendix A1, A2, A3, A4 (as shown at the end of this rebuttal)). Bandwidths used are 0.2 and 0.3 as opposed to 0.1 used previously. Given the resolution of the KDEs, class differences should have a minimal impact on the distributions, especially when the amount of data points is large enough. For the higher rain rates with fewer data points, yes, probably, the difference class definition would have an impact on the KDEs.

Here below is the new Figure 5:

[Figure]

26. μ and μ0 are used interchangeably here. It would be of interest to show the mean DSD per instrument (ie mean/median and bars for quantiles on N(D) by D class) to empirically show the differences in shape.

Response: μ should have been the only notation used here. These are typo that remains from a initial version of the manuscript where μ$_0$ was used instead.

The mean/median of N(D) by D will be very similar or identical to our current Figure 3, where we plot the total number of recorded droplet by D (common overlapping bins for the two instruments). In the manuscript, we already explore in depth the variability of the recorded droplets per bin size with the KDEs plots. Adding the quantiles for N(D) would rather be anecdotic additional information.

27. Lines 357–358: "one of the Thies LPM instruments (T3) measured a very large number of particles falling into these two categories" – I do not see this very large number of particles in Figure 6, which shows more particles outside the expected velocity ranges for OTT1 and OTT3.

Response: Figure 6 is now updated showing also the non-filtered particles (outside of the boundaries defined using Atlas et al. (1973) model of fall velocity). Our apologies for this mistake: the OTT figure was the correct version showing non-filtered and filtered while the Thies LPM were showing only the filtered particles. Figure 6 now supports this statement as you can see for T3 in particular.

28. Figure 7: Please clarify "mean diameter" as meaning DSDs with small drops removed, since "mean diameter" could be taken to discussion on lines 369–373 is confusing and requires rephrasing – is the point that the Thies instruments seem to have a lot of drops in the first class after 0.6 mm, where as the drops are more evenly distributed in the OTT cases?

Response: it now reads "and for minute data **meeting $D_m > 0.6$ mm** (red)." Maybe the explanation given by the reviewer is a way to interpret the difference, but we think it would be speculative to interpret the differences in the distributions of drop per bin size… as there are many factors that can explain this: sensitivity of the instruments, differences in threshold sensitivities, manufacturer post-processing software…etc

29. Lines 374–375 and Figure 7: I interpret the plot for R in Figure 7 as showing that without the small drops there are more very low rain rates, since for example the left tail on the solid red line is left of the blue and green lines; this appears to clash with the statements made in the article text. In the OTT distributions and the T1 > 0.6 distribution there are rain rate values less than 0.1 which was stated to be the minimum allowed in these analyses. What explains these low values?

Response: Thanks very much for noticing this issue. We looked back in our code, and in the version of figure 7 we included in the manuscript the data was indeed also including rain rates < 0.1 mm/h. We have now corrected the figure such that it includes only the rain rates > 0.1 mm/h, as indicated in the manuscript method section. We revised the last sentence of the paragraph introducing Figure 7, especially in regards to the impact of the filtering on the instruments integrated variables. It now reads:

"Only the fitting parameters $N_w$ and $\mu_0$ as well as $D_m$ were slightly affected for the Thies LPM. In contrast, the OTT Parsivel[1] data were less affected, as expected since the OTT Parsivel[1] recorded smaller amounts of droplets with diameter ranges < 0.6 mm."

Here below is our new Figure 7:

[Figure]

30. Lines 391–392: The statement here (that differences between instruments are larger as R increases) is not supported by Figure 4, in which there is not a clear effect on the differences that correlates with the peaks in R.

Response: We have removed this sentence. The paragraph now starts directly with the subsequent sentence, which introduces the figures in the appendix and Figure 8, exploring the effect of rainfall rate on the integrated variables.

31. Lines 397–398: The statements here are not supported by the plots in Figure 8. OTT1 shows similar frequencies to OTT3 for for high reflectivity, attenuation, and R. The big difference is that in the OTT1 distributions there are more low values and less frequent mid-range values than in OTT3 distributions.

Response: We agree and have changed the sentence to:

"OTT1 showed similar frequencies to OTT3 for rain rates, reflectivity and attenuation values, but in OTT1 there were more low values and less frequent mid-range values than in OTT3 distributions. Both of the Thies LPM statistics were similar."

32. Line 398: Which variables are meant by "first order" moments here? Since high rain rates mean many drops of all sizes, the stated link between high rain rates and large-drop sampling uncertainty requires some more argument, e.g. by looking specifically at the variables influenced more by large drop occurrences Z). I think that sampling uncertainty due to sample numbers decreasing with increasing rain rates may play a much larger role than the large drops in the observed differences (e.g. there are only 129 points in Figure 8, but 29815 in Figure A1).

Response: Thanks: we have removed the terminology "first order" which was confusing. We explain in the text that the sampling effect is a factor, and we added the effect of the small sample size. It now reads:

"All DSD parameters (Figure 8) started to show discrepancies between all instruments for rain rates > 10 mm h$^{-1}$, due to the sampling effect related to the occurrence of larger drops falling erratically in space and time, therefore being captured by some instruments while not by co-located neighbours, this being enhanced by the small data sample (128 minutes of data in Figure 8). "

33. Lines 415–419: It is not clear why it is important to separate 0.6 mm, or which previous bimodal distributions the authors are referencing. A reference to the scheme used to separate convective from stratiform regimes should be included here.

Response: Thanks, we added more information on the bimodal distributions and on the scheme used to separate convective and stratiform rainfall. It now reads:

"Figure 9 shows the scatter plots together with fitted $Z_H$-R relations, with the fitting for DSD done corresponding to $D_m < 0.6$ mm and to $D_m > 0.6$ mm. Indeed, in the frequency distributions seen in the previous sections, integrated parameters of the DSD showed the occurrence of bimodal distributions (for $N_W$, $D_0$ and $D_m$ in particular). This implies that if considering the full dataset, there should be at least two corresponding power-law relations for each distribution of the data. We have

now re-defined the categories as: convective rainfall ($Zh > 30$ dBZ), stratiform rainfall ($Zh < 30$ dBZ and $D_m > 0.6$ mm) and drizzle ($D_m < 0.6$ mm). These are shown in Table 3."

Here below is our new Figure 9:

[Figure]

34. Lines 428–430: The statement that b decreases with raindrop size diameter increases is not true when comparing the fits on the two stratiform data sets, and because all data contains the convective data it is hard to compare the results for "all" to the convective results.

Response: We realised that our definition of stratiform – and having two stratiform labelled data categories was confusing, and the likely cause of your comment. We have now re-defined the categories as: convective rainfall ($Zh > 30$ dBZ), stratiform rainfall ($Zh < 30$ dBZ and $D_m > 0.6$ mm) and drizzle ($D_m < 0.6$ mm). When comparing stratiform rainfall to convective rainfall, the statement "b decreases with raindrop size diameter increases" is correct as demonstrated in the literature to date. We then discuss the case of drizzle separately in the next paragraph.

Line 444: While true that this paper showed Thies can capture small drops related to drizzle, no estimate of the measurements' accuracy can be made without another instrument that also captures those drop sizes.

Response: Indeed: we added an additional sentence to clarify that aspect. The new sentence reads:

"The findings of this paper showed that the Thies LPM has the capacity to capture this part of the DSD spectrum, although some additional research using co-located disdrometers also capturing this lower part of the DSD spectrum will be needed to evaluate the accuracy of these Thies LPM measurements."

35. Table 4: log → were these differences calculated; i.e. for each value in the left-most column, what was the class size in dBZ? Which instrument was taken as the reference, i.e. the percentage is of which value of R?

Response: T1 was taken as the reference. We have now added this information in the table caption.

36. Figure 9: The logarithms should be the axis labels. Some brief discussion in the main text about the differences between the fitted relationships and the normal Marshall-Palmer Z-R relation should be included. Also, the b exponent in the Marshall-Palmer version is 1.6 not 1.5 as stated in the plot key; please include a reference to Marshall 1955 in the caption for this plot and clarify which relationship is shown in the plot.

Response: Reference to Marshall et al. (1955) has been added as well as the corresponding relationship in the caption of the figure. We corrected the figure box legend, with b exponent of the MP relation equal to 1.6 as well as the plotted relation on the figure (which was mistakenly plotted initially with b = 1.5). The Figure now doesn't mention the logarithms in the axis legend but specify the units of the legends as dBZ and dBR for the y- and x- axis.

We added two sentences to compare to Marshall Palmer:

(1) At the end of the first paragraph of section 3.7: "Relationships considering all data were the closest to the Marshall-Palmer relations, and differed significantly for stratiform rainfall for both instrument types."
(2) In the 4th paragraph of the same section, last sentence now reads: "Disdrometer-derived $Z_H$-$R$ relations as compared to the Marshall-Palmer relation $Z_H$ =200R$^{1.6}$ led to a bias in rainfall rates for reflectivities of 50 dBZ of up to 21.6 mm h$^{-1}$."

37. Figure 11 caption: "augmented by numbers of authors since" – which authors? Please provide references.

Response: We added Marzuki et al. (2013) to the reference list, in which the authors summarise in a plot the additional observations to Bringi et al. (2003).

38. Line 544: Another possible future direction could be comparisons of Thies LPM to other instruments that, unlike Parsivel, are able to accurately measure concentrations of small drops.

Response: Thanks! We added this additional sentence to the manuscript after the first proposed research direction.

39. Line 545: The reference to Raupach et al. 2019 has year 2019 in the references list but 2018 in the text.

Response: It has now been corrected to Raupach et al. (2019) in the text.

40. Lines 525–535: The appearance and discussion of Figure 11 do not fit well into this paragraph, which is ostensibly about a lack of DSD observations in Australia. Are the authors aiming to highlight the mismatch between their observations and the climate regimes shown in Figure 11? This discussion feels incomplete.

Response: We have split that original paragraph into two distinct ones: the first one starts with the original: "For the first time…" and discuss the novel observations in that climatological context.
* * *
New appendix Figures as per below:

[Figure]

Revised Figure A1 above

[Figure]

Revised Figure A2 above

[Figure]

Revised Figure A3 above

[Figure]

Revised Figure A4 above

---

## Author Response (AR2)

**Response to the Editor**
*Guyot et al.* **under review at HESS**

**https://doi.org/10.5194/hess-2019- 277, 2019**

We would like to thank the editor for his comments and the notes on our manuscript. We have taken into account all of his suggestions and have prepared a final manuscript.

Tracked changes on the manuscript can be seen in the pages below.

On behalf of the co-authors,
Yours sincerely,

Dr Adrien Guyot
Monash University

[revised manuscript text omitted]